# Go Beyond Earth: Understanding Human Actions and Scenes in Microgravity Environments

**Di Wen**[1][*]   **Lei Qi**[1][*]   **Kunyu Peng**[1][†]   **Kailun Yang**[2]   **Fei Teng**[2]   **Ao Luo**[3]   **Jia Fu**[4,5]
**Yufan Chen**[1]   **Ruiping Liu**[1]   **Yitian Shi**[1]   **M. Saquib Sarfraz**[1]   **Rainer Stiefelhagen**[1]
[1]Karlsruhe Institute of Technology   [2]Hunan University   [3]Waseda University
[4]KTH Royal Institute of Technology   [5]RISE Research Institutes of Sweden

## Abstract

Despite substantial progress in video understanding, most existing datasets are limited to Earth's gravitational conditions. However, microgravity alters human motion, interactions, and visual semantics, revealing a critical gap for real-world vision systems. This presents a challenge for domain-robust video understanding in safety-critical space applications. To address this, we introduce MicroG-4M, the first benchmark for spatio-temporal and semantic understanding of human activities in microgravity. Constructed from real-world space missions and cinematic simulations, the dataset includes 4,759 clips with 13,261 action annotations covering 50 actions, 1,238 context-rich captions, and over 7,000 question–answer pairs on astronaut activities and scene understanding. MicroG-4M aims to support three core tasks: fine-grained multi-label action recognition, temporal video captioning, and visual question answering, thereby enabling a comprehensive evaluation of both spatial localization and semantic reasoning in microgravity contexts. We establish baselines using state-of-the-art models. All data, annotations, and code are available at https://github.com/lei-qi-233/MicroG-4M.

## 1 Introduction

Yuri Gagarin's historic flight in 1961 marked the beginning of human space exploration. Since then, significant milestones have been achieved, including crewed lunar landings, the continuous operation of the International Space Station (ISS) for over 25 years, and the participation of more than 650 individuals in space missions Moskowitz & Wolf (2025). With numerous planned crewed missions in the near future, the frequency and complexity of human activities in space are expected to increase substantially. In this context, ensuring the safety, enhancing the operational efficiency of space missions, and safeguarding the health and well-being of astronauts are of paramount importance. With the rapid advancement of artificial intelligence, the integration of robotic systems aboard spacecraft is anticipated in the near future. These systems will assist astronauts in routine and mission-critical tasks. Consequently, there is a growing demand for the development of deep learning-based scene understanding and action recognition methods tailored specifically for the unique challenges posed by the microgravity environment.

Human action recognition Wang et al. (2023a); Feichtenhofer (2020); Feichtenhofer et al. (2019); Fu et al. (2019), video captioning Chen et al. (2024), and Visual Question Answering (VQA) Hu et al. (2023) are essential for intelligent human-robot collaboration, particularly in space, where precise perception and understanding of astronauts' actions and their surrounding context are crucial for ensuring operational safety, efficiency, and autonomous assistance under constrained conditions. This capability is crucial for ensuring mission efficiency, enhancing astronaut safety, and providing autonomous assistance in the confined and complex conditions of space habitats. Numerous human action recognition datasets and video caption datasets have been developed, playing an important role in various research areas within computer vision and in industrial applications Camarena et al. (2023);

---

[*]Equal contribution.
[†]Corresponding author: kunyu.peng@kit.edu

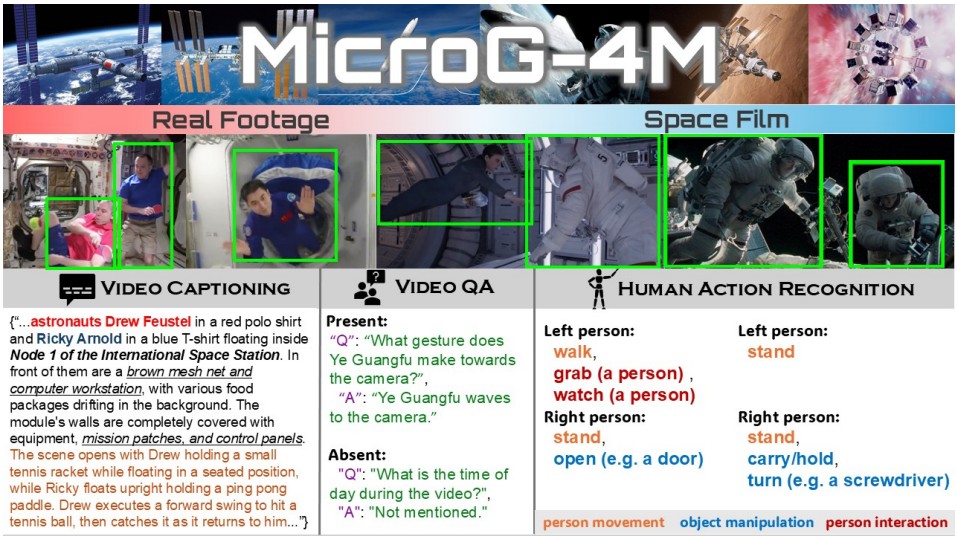

Figure 1: An illustration of the MicroG-4M, containing videos from real and simulated microgravity environments (*e.g.*, movies). The dataset supports benchmarks for three tasks: (1) video captioning, (2) video question answering, and (3) fine-grained human action recognition under microgravity.

Pareek & Thakkar (2021); Al-Faris et al. (2020); Le et al. (2022). However, most of the existing datasets for video captioning and action recognition are recorded on Earth without microgravity settings, *e.g.*, Kinetics400 Kay et al. (2017), AVA Gu et al. (2018), FineGym Shao et al. (2020), EgoCross Li et al. (2025), and ActivityNet Captions Chen et al. (2019).

Human actions in space depart markedly from their terrestrial counterparts because microgravity removes gravity-aligned orientation and reliable support surfaces. Basic behaviors—standing, locomotion, eating, and manual manipulation—follow different kinematics and contact patterns Hagio et al. (2022). Standing becomes orientation-invariant, often maintained via foot restraints or handholds rather than ground support; locomotion is achieved by drifting or pulling along structures rather than gait; and manipulation frequently involves releasing or catching free-floating objects instead of placing or picking from a surface. These shifts violate terrestrial orientation, support/contact, and object-dynamics priors, which helps explain the degradation of Earth-trained Human Action Recognition (HAR) models in orbit and motivates a microgravity-specific benchmark.

To address this gap, we introduce *MicroG-4M*, a new video benchmark specifically designed for spatio-temporal and semantic understanding of human activities in microgravity. The name "4M" reflects four characteristics: *Multi-source* (Real mission footage and physically plausible film), *Multimodal* (RGB + text annotations), *Multi-task* (HAR, captioning, VQA), and *Microgravity*. MicroG-4M comprises 4,759 three-second video clips drawn from public YouTube footage of real space missions and carefully selected, realistic space-themed films to augment scenario diversity and coverage. The clips span scenes inside the ISS, the Tiangong Space Station, crewed spacecraft cabins, and extravehicular activities. The corpus contains more than 390,000 annotated frames with bounding boxes for each visible individual and 13,000+ action labels covering 50 distinct actions. In addition, it includes 1,238 descriptive clip captions created by human annotators and 7,000+ open-ended question–answer pairs aimed at assessing factual, causal, and counterfactual understanding of scenes in microgravity. Both captions and QA annotations were carefully curated through multiple refinement rounds and data selection stages, ensuring high semantic fidelity, contextual accuracy, and relevance to the microgravity setting. Together, these resources enable multi-label spatio-temporal detection, fine-grained action recognition, caption generation, and VQA within a single dataset. To evaluate existing methods, we build the first comprehensive benchmark for these tasks, dubbed **MicroG-Bench**. We evaluate representative video encoders (SlowFast Feichtenhofer et al. (2019), X3D Feichtenhofer (2020), and MViT Li et al. (2022)) for human action recognition and leading vision-language models (*e.g.*, InternVideo Wang et al. (2022), Gemini 1.5 Pro Reid et al. (2024), GPT-4o Hurst et al. (2024)) for captioning and VQA tasks. Across all evaluated subtasks, state-of-the-art terrestrial models experience significant performance degradation, underscoring the unique

challenges posed by microgravity environments, such as arbitrary orientations, floating objects, and confined spacecraft interiors. These results highlight the necessity of dedicated benchmarks to facilitate the advancement of more robust and generalizable AI systems tailored for space applications. MicroG-Bench, therefore, provides a unified, fair yardstick that will guide the development of more robust and generalizable perception and language systems for astronaut assistance and autonomous mission operations.

We summarize our contributions as follows. We introduce **MicroG-4M** and **MicroG-Bench**, establishing the **first comprehensive benchmark** for microgravity video understanding. Beyond providing 4,759 richly-annotated clips across 50 actions, we define a rigorous evaluation protocol with standardized splits for HAR, Captioning, and VQA. Crucially, our in-depth empirical analysis provides the core insight: by benchmarking strong baselines, we quantify a persistent performance gap and uncover characteristic failure modes unique to weightlessness. These findings expose the limitations of terrestrial models, positioning MicroG-4M as the **foundational standard** for developing future microgravity-aware architectures.

## 2 RELATED WORK

**Vision-Based Understanding in Microgravity and Space Environments.** Microgravity challenges terrestrial vision assumptions, requiring perception models for space habitats. Early work showed traditional SLAM to be unreliable, leading to robust alternatives with visual-inertial fusion, semantic mapping, CAD-informed constraints Soussan et al. (2022); Mao et al. (2024); Miller et al. (2022); Tweddle et al. (2015), and validated on platforms like Astrobee Kang et al. (2024). Real-time vision has also enabled dynamic tasks such as target tracking and scene change detection Oestreich et al. (2021); Dinkel et al. (2024). Research has shifted toward interaction, including astronaut pose recovery Gan et al. (2023); Ouyang et al. (2025), gesture recognition Lingyun et al. (2020); Gao et al. (2020), and EMG-based input Assad et al. (2013). Yet, high-level semantic understanding of astronaut actions and intent remains limited. Current assistants like CIMON Eisenberg et al. (2024) provide basic perception but lack contextual reasoning. To address this, we propose a unified benchmark for action recognition, video captioning, and visual question answering in space operations.

**Datasets for Action Detection, Video Captioning, and VQA.** Progress in video understanding has been largely enabled by the introduction of benchmark datasets across action detection, captioning, and VQA. For action detection, early datasets such as UCF101 Soomro et al. (2012) and HMDB51 Kuehne et al. (2011) provided trimmed classification tasks, later extended by Kinetics Kay et al. (2017) and ActivityNet Caba Heilbron et al. (2015) to large-scale, untrimmed settings with diverse action categories. AVA Gu et al. (2018) further introduced spatio-temporal annotations for atomic actions, enabling fine-grained multi-label detection in realistic scenes. Video captioning evolved from MSVD Chen & Dolan (2011) and MSR-VTT Xu et al. (2016), which paired short clips with multiple sentences, to dense captioning in untrimmed videos via ActivityNet Captions Krishna et al. (2017) and procedural datasets like YouCook2 Zhou et al. (2018). Multilingual benchmarks such as VATEX Wang et al. (2019) expanded the task to cross-lingual settings with high-quality parallel annotations. In VQA, static image datasets like VQA v2.0 Goyal et al. (2017) and CLEVR Johnson et al. (2017) laid the foundation for compositional reasoning, while TGIF-QA Jang et al. (2017), MovieQA Tapaswi et al. (2016), and TVQA Lei et al. (2018) introduced temporal and multimodal reasoning in video. Recent efforts such as NExT-QA Xiao et al. (2021) and CLEVRER Yi et al. (2020) focus on causal inference and counterfactual reasoning. Existing datasets focus on terrestrial settings and lack the unique complexity of space activities. To address this, we introduce the first microgravity benchmark for video captioning and VQA, enabling evaluation in long-horizon, space-relevant scenarios.

**Fine-Grained Human Action Recognition.** Fine-grained video-based action recognition Gritsenko et al. (2023); Chung et al. (2021) aims to detect fine-grained, indivisible human actions in both single- and multi-person videos. Unlike standard clip-level recognition, atomic action localization requires frame-level, multi-label classification with spatio-temporal bounding box predictions. To tackle this, CNN-based Wang et al. (2023a); Feichtenhofer (2020); Feichtenhofer et al. (2019) and transformer-based Ryali et al. (2023); Wang et al. (2023a); Li et al. (2022); Wang et al. (2023b; 2022); Peng et al. (2022); Gritsenko et al. (2023) models have been adapted by adding multi-label heads, bounding box regressors, and region-of-interest modules. Notably, Ryali *et al.*Ryali et al.

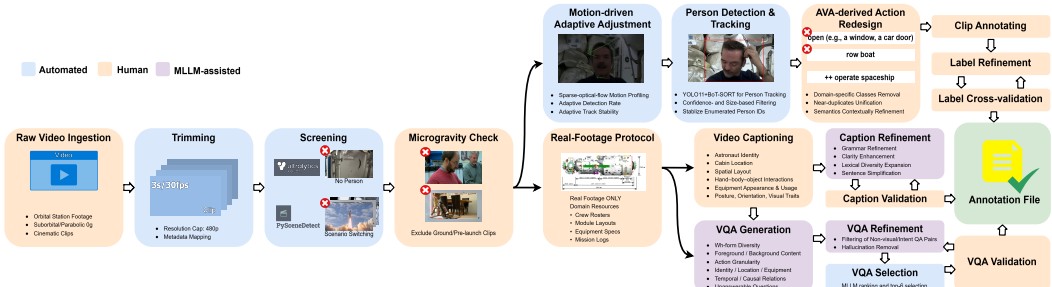

Figure 2: Overview of the MicroG-4M collection and annotation pipeline. Blue, orange, and purple modules denote automated, human, and MLLM-assisted stages. Raw footage is trimmed into 3-second clips, automatically filtered, and manually verified to retain authentic microgravity scenes. Motion-driven detection produces person-level boxes for HAR, while a real-footage protocol constrains captioning and VQA to genuine microgravity content. AVA-based action labels, along with caption and VQA generation and validation, produce the final annotations for action recognition, captioning, and VQA.

(2023) introduced a hierarchical vision transformer balancing accuracy and efficiency, while Wang *et al.* Wang et al. (2023a) proposed a dual-masked autoencoder for improved video pretraining. In this work, we evaluate such HAR methods for the first time in microgravity scenarios.

# 3 COLLECTION METHODOLOGY

We aim to construct a dataset comprising authentic footage recorded aboard the International Space Station and other spacecraft, as well as films that realistically depict microgravity conditions. In the following, we introduce our comprehensive pipeline for collecting, assembling, filtering, and annotating the MicroG-4M dataset, which supports multi-label spatio-temporal action detection, video captioning, and video question answering tasks from Internet-sourced videos. An overview of this collection and annotation pipeline is shown in Figure 2.

**Raw Video Information Collection.** Video sources primarily include genuine microgravity footage from actual spacecraft missions and selected cinematic clips known for their realistic depictions of weightlessness. Authentic spacecraft videos were mainly retrieved from online video platforms, while cinematic clips were manually chosen based on their fidelity to real microgravity conditions. Videos were downloaded with a target resolution of 480p whenever available, and otherwise at the highest available resolution below 480p. The final dataset comprises approximately 5,000 three-second clips, predominantly composed of authentic space station footage.

**Dataset Assembly Pipeline.** We established an automated pipeline to preprocess and structure raw videos, enabling consistent and accurate downstream annotation. The pipeline consists of three main stages: video segmentation, filtering, and automated bounding-box annotation.

Raw Video Trimming. Raw videos are trimmed into uniform three-second clips at 30 fps, discarding shorter segments to maintain temporal consistency.

Filtering. Video clips undergo automatic filtering based on person detection and scene-transition analysis. Specifically, we employed YOLOv11 Khanam & Hussain (2024) for human detection and PySceneDetect Castellano to identify abrupt scene changes, discarding clips with insufficient human-action content or disrupted temporal continuity.

Bounding-box Annotation. Person bounding boxes are automatically annotated using YOLOv11 Khanam & Hussain (2024) detection combined with BoT-SORT tracking Aharon et al. (2022). To enhance annotation accuracy, adaptive strategies were employed by assessing video motion intensity using sparse optical flow methods Jeannin & Divakaran (2001); Ali (2013); Szeliski (2010). Annotation parameters were dynamically adjusted accordingly to optimize computational efficiency and annotation precision. The resulting structured annotations include bounding-box coordinates, unique identities, and detection confidence scores.

**Manual Video Screening.** Following automated preprocessing, an additional manual verification step was performed to ensure dataset purity and environmental consistency. Specifically, each generated video clip was individually reviewed to exclude terrestrial scenes, such as ground-based footage or pre-launch preparations, thereby retaining only those segments clearly depicting human activities under authentic microgravity conditions. This rigorous screening process ensured semantic precision and enhanced the overall reliability of the dataset.

**Action Label Annotation.** In this phase, we derive a microgravity-tailored action taxonomy from AVA's 80 atomic actions Gu et al. (2018). To ensure environmental applicability and semantic continuity, we (i) exclude actions that are physically inapplicable in space (*e.g.*, water- or ground-specific), (ii) merge near-duplicates into unified categories, and (iii) introduce context-aware semantic adjustments. To disambiguate visually or semantically similar actions, we define explicit differentiation criteria that standardize annotation decisions. Each three-second clip is treated as a self-contained unit: for every detected individual, annotators assign up to five visible or inferable action labels per clip. The verified annotations are exported as structured CSV files for downstream training and evaluation. Retaining AVA class names while re-grounding their semantics in microgravity enables fair Earth→space comparisons without altering the label space.

**Caption and VQA Annotation.** For caption and VQA annotation, annotators consulted publicly available, authoritative resources, including official crew rosters and astronaut biography databases from major space agencies (*e.g.*, NASA, CNSA, ESA), historical spacecraft module layout diagrams, equipment specification documents, mission operation reports and task logs, and onboard video transcripts and archival footage. These domain resources are systematically used to resolve ambiguities in astronaut identity, cabin context, and equipment usage, and to enforce consistent naming conventions for modules and objects.

For caption annotation, captions are written by human annotators based on frame-by-frame visual inspection of each 3-second clip at 30 fps. Beyond the core action and object description, captions explicitly encode astronaut identity, cabin location and background spatial layout, fine-grained hand–body–object interactions, equipment appearance and usage, and posture, orientation, and other distinctive visual traits (*e.g.*, helmet and suit state). All factual content is cross-checked against the domain resources above to ensure semantic accuracy. Multimodal Large Language Models (MLLMs) are applied in a subsequent refinement step to improve grammatical fluency, clarity, lexical diversity, and to simplify overly complex sentences into a standard captioning style; all edited captions are re-validated by annotators. These high-quality captions provide strong semantic grounding to support downstream tasks such as action recognition and context-aware retrieval.

For Video Question Answering (VQA), we adopt a caption-grounded, two-stage generation and filtering pipeline guided by Heilman & Smith (2009). Given each refined caption, candidate question–answer pairs are generated via MLLMs using prompt templates designed to elicit variation across multiple reasoning dimensions, including standard Wh-forms, foreground versus background elements, coarse versus fine-grained actions, identity, location and equipment, and temporal or causal relations. One deliberately unanswerable question per clip is optionally included. A refinement pipeline then filters out candidates that rely on sound, latent intent, or non-visual cues. The remaining QA pairs are ranked by the MLLM based on logical consistency, linguistic fluency, semantic relevance, and information value. Annotators review the top-ranked pairs, remove any hallucinated or fabricated content, revise prompts if necessary, and verify that all retained questions are visually grounded or explicitly marked as "Not mentioned" when unanswerable. To ensure comprehensive coverage from broad contextual understanding to detailed actions while maintaining visual fidelity and a controlled reasoning scope, a final set of six diverse QA pairs is retained for each clip.

**Annotation Quality Control.** During the annotation phase, a team of 9 annotators collaboratively worked on labeling the fine-grained human actions, generating video captions, and crafting VQA pairs. To ensure high annotation quality, a comprehensive cross-verification protocol was implemented throughout the process, supported by group discussions to resolve disagreements and reach consensus. For the captioning and VQA tasks, all entries were further subjected to a semantic consistency check utilizing LLMs, followed by iterative human review to enhance both linguistic clarity and factual accuracy. This layered validation pipeline, combining automated and manual strategies, ensured the reliability and coherence of the annotations across all modalities. As a result, we introduce the MicroG-4M dataset, an extensively curated benchmark explicitly designed for fine-grained, multi-label, spatiotemporal action recognition, captioning, and VQA under microgravity conditions. This

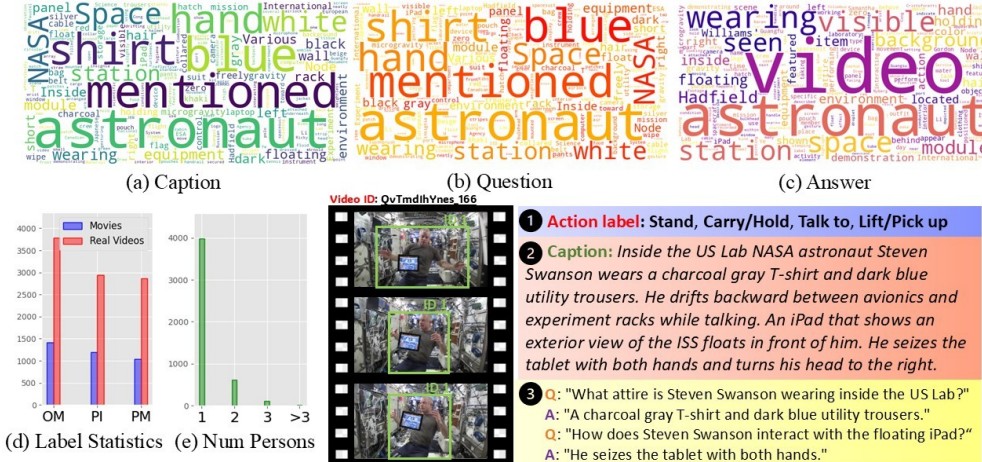

Figure 3: An illustration of the statistics of the dataset and the annotation samples. Word clouds of the (a) Caption, (b) Question, and (c) Answer from our MicroG-4M dataset are provided. The label statistics of the fine-grained human action recognition are provided in (d), which showcases the annotation number per action group (*i.e.*, Object Manipulation (OM), Person Interaction (PI), Person Movement (PM)). The distribution of person counts per video clip is visualized in (e). On the bottom right, one annotation sample from MicroG-4M is provided.

dataset sets a new foundation for advancing robust scene understanding and human activity analysis in space-based environments.

# 4 DATASET COMPOSITION

We release the *MicroG-4M* dataset, comprising the following subsets: **Fine-Grained Human Action Recognition Subset:** Contains fine-grained, multi-label annotations for spatiotemporal human action detection, comprising 4,759 manually annotated three-second video clips from authentic microgravity environments. Each clip includes annotations for up to five distinct actions per detected individual. The 50 action labels are organized into three categories: Object Manipulation (4,986 annotations, 37.60%), Person Interaction (4,288 annotations, 32.34%), and Person Movement (3,987 annotations, 30.07%). Object Manipulation includes actions involving physical objects or equipment, Person Interaction involves social or physical engagement with others, and Person Movement covers individual postural or locomotor motion. These categories are derived from AVA's Gu et al. (2018) atomic action labels and regrouped for semantic coherence under microgravity conditions, following macro-level groupings similar to those adopted in AVA-based fine-grained action recognition benchmarks Peng et al. (2024). Figure 3(d) illustrates the distribution of these categories in both real and simulated video clips, while Figure 3(e) shows the distribution of person counts per clip, highlighting the diversity of social configurations. Overall, the subset contains 13,261 action annotations, including 9,610 from real footage, 3,651 from simulated sources, and 390,000 bounding box annotations. **Video Caption Subset:** Comprises 1,238 detailed, semantically rich descriptions validated against official aerospace agency documentation for astronaut identities, spacecraft locations, actions, appearances, and interactions. Each caption corresponds uniquely to one video clip. The word cloud visualization in Figure 3(a) provides insights into frequently mentioned terms and concepts, illustrating the thematic and semantic distribution of the captions. **Visual Question Answering (VQA) Subset:** Includes 7,428 structured QA pairs, systematically generated and refined via MLLMs to ensure linguistic fluency, semantic relevance, and comprehensive coverage of detailed actions and broader context, with each of the 1,238 video clips associated with up to six diverse QA pairs. Figure 3(b) and (c) show word clouds for questions and answers separately, revealing prevalent inquiry types and common semantic patterns within the dataset. Figure 3 presents representative video clips from the dataset, demonstrating typical annotation examples and highlighting the visual and semantic diversity of the dataset.

## 5 EXPERIMENTS VALIDATING

### 5.1 BENCHMARK PROTOCOL

**Data Split.** We partition the dataset into training, validation, and test subsets in a 7:1:2 ratio. Row-level distribution is as follows: Training set: 9,273 records (69.93% of 13,261); Validation set: $1,330$ records (10.03% of 13,261); Test set: 2,658 records (20.04% of 13,261). Video-level distribution is as follows: Training set: 3,331 videos (69.99% of 4,759); Validation set: 475 videos (9.98% of 4,759); Test set: 953 videos (20.03% of 4,759).

**Baselines for Fine-Grained HAR.** For fine-grained action recognition, we evaluate well-established baselines from video-based human action recognition, including transformer-based models (MViTv1 Fan et al. (2021), MViTv2 Li et al. (2022)) and CNN-based models (I3D Carreira & Zisserman (2017), SlowFast Feichtenhofer et al. (2019), X3D Feichtenhofer (2020), C2D Feichtenhofer et al. (2019)). These widely used architectures cover both paradigms and allow us to assess generalization to the unique spatiotemporal dynamics of microgravity. All models are initialized with Kinetics400 Kay et al. (2017) pretrained weights for better convergence. HAR experiments were conducted on NVIDIA RTX 2080 Ti GPUs and an AMD Ryzen Threadripper 2990WX CPU.

**Baseline methods for Video Captioning and QA.** For video captioning and question answering in microgravity, we evaluate strong baselines, including open-source models (VideoChatGPT Li et al. (2023), mPLUG-Owl-3 Ye et al. (2025), LLaVA-Next Li et al. (2024), VideoLLaVa Lin et al. (2024), Qwen-2.5-VL Bai et al. (2025), InternVideo Wang et al. (2022)) and closed-source models (GPT-4o Hurst et al. (2024), Gemini 1.5 Pro Reid et al. (2024), Gemini 2.5 Pro Comanici et al. (2025)). Open-source models offer reproducibility via public weights and code, while closed-source models serve as upper-bound references. This mix enables both transparent analysis and comprehensive evaluation of video-language understanding in microgravity. All evaluations were conducted using NVIDIA A100-40 GPUs and Intel Xeon Platinum 8368 CPUs.

**Cross-domain transfer protocol.** To quantify the microgravity-induced domain gap, we use a fixed transfer setup: models pretrained on Kinetics are fine-tuned on AVA Gu et al. (2018) with matched settings, then evaluated zero-shot on MicroG-4M. For terrestrial contrast, we test on JHMDB (Split 1) Jhuang et al. (2013) with the overlapping action set and standard protocol. This isolates domain effects (microgravity vs. Earth) from implementation choices; evaluation details in Sec. 5.

**Evaluation Metrics for Fine-Grained HAR.** Our evaluation metrics include mAP@0.5, F1 score, recall, and AUROC, all calculated using the macro method. Among these, mAP@0.5 is the primary metric for measuring average detection accuracy per category and thus comprehensively evaluating the model's action recognition performance in a microgravity environment. Per-class threshold sweeps provided in Appendix.

**Evaluation Metrics for Video Caption and QA.** We adopt standard automatic evaluation metrics, including CIDEr Vedantam et al. (2015), BLEU-4 Papineni et al. (2002), ROUGE-L Lin (2004), METEOR Banerjee & Lavie (2005), and BERTScore (F1) Zhang et al. (2020), all rescaled to a 0∼100 range for consistency. For semantic similarity, we report S-BERT Reimers & Gurevych (2019a) and S-VQA Pathak et al. (2023) scores, both computed as cosine similarity between Sentence-BERT Reimers & Gurevych (2019b) embeddings of predicted and reference texts. S-VQA is used specifically for answer evaluation in generative visual question answering settings, capturing semantic equivalence beyond lexical overlap. More details of the implementations are delivered in Appendix.

### 5.2 RESULT ANALYSIS FOR FINE-GRAINED HUMAN ACTION RECOGNITION

**Quantitative Analysis.** We evaluate several representative models on MicroG-4M, all pretrained on Kinetics400 Kay et al. (2017) and fine-tuned on our dataset. As shown in Table 1, results plateau around 47% test mAP, indicating a substantial gap to Earth-trained regimes. The ranking also inverts common trends on Kinetics/AVA, with CNNs plus non-local modules Feichtenhofer et al. (2019); Carreira & Zisserman (2017) leading mAP/AUROC and Slow (4×16) Feichtenhofer et al. (2019) yielding the best F1, suggesting that local spatial encoding and structured receptive fields remain advantageous when motion lacks gravitational consistency. Longer temporal windows further help, highlighting the need for domain-adapted temporal reasoning. Under the matched transfer protocol, AVA→MicroG-4M underperforms AVA→JHMDB (Table 3), isolating a physics-driven

Table 1: Performance of models fine-tuned on MicroG-4M, evaluated on the validation and test sets.

| Model | | | | Validation | | | | Test | | | |
|---|---|---|---|---|---|---|---|---|---|---|---|
| Arch | TC | Backbone | #Params (M) | mAP (%) | F1-score (%) | Recall (%) | AUROC (%) | mAP (%) | F1-score (%) | Recall (%) | AUROC (%) |
| C2D Feichtenhofer et al. (2019) | 8x8 | R50 He et al. (2016) | 23.61 | 27.22 | 12.52 | 10.34 | 82.86 | 29.51 | 8.09 | 6.58 | 83.49 |
| C2D NLN Feichtenhofer et al. (2019) | 8x8 | R50 He et al. (2016) | 30.97 | 40.42 | 23.10 | 20.41 | 87.11 | 44.64 | 28.30 | 24.86 | 89.40 |
| I3D Carreira & Zisserman (2017) | 8x8 | R50 He et al. (2016) | 27.33 | 40.93 | 19.78 | 16.93 | 86.44 | 46.37 | 26.37 | 22.25 | 88.79 |
| I3D NLN Carreira & Zisserman (2017) | 8x8 | R50 He et al. (2016) | 34.68 | 41.42 | 24.11 | 23.00 | 86.37 | 47.12 | 28.07 | 24.65 | 88.52 |
| Slow Feichtenhofer et al. (2019) | 8x8 | R50 He et al. (2016) | 31.74 | 40.32 | 21.83 | 19.08 | 84.55 | 45.19 | 26.13 | 22.77 | 88.49 |
| Slow Feichtenhofer et al. (2019) | 4x16 | R50 He et al. (2016) | 31.74 | 42.97 | 22.73 | 19.71 | 85.46 | 46.37 | 28.72 | 25.38 | 88.30 |
| SlowFast Feichtenhofer et al. (2019) | 8x8 | R50 He et al. (2016) | 33.76 | 38.76 | 20.29 | 17.66 | 85.91 | 43.02 | 22.63 | 18.98 | 88.51 |
| SlowFast Feichtenhofer et al. (2019) | 4x16 | R50 He et al. (2016) | 33.76 | 37.10 | 17.74 | 14.90 | 84.94 | 42.09 | 23.69 | 20.18 | 87.54 |
| MViTv1 Fan et al. (2021) | 16x4 | B-CONV | 36.34 | 17.79 | 7.89 | 6.86 | 72.40 | 12.86 | 5.54 | 4.66 | 74.63 |
| MViTv2 Li et al. (2022) | 16x4 | S | 34.27 | 17.57 | 8.31 | 6.92 | 72.67 | 15.14 | 8.16 | 7.17 | 78.61 |
| X3D Feichtenhofer (2020) | 13x6 | S | 2.02 | 17.59 | 6.63 | 5.63 | 78.27 | 14.07 | 5.77 | 4.52 | 78.23 |
| X3D Feichtenhofer (2020) | 16x5 | L | 4.37 | 23.56 | 8.82 | 7.38 | 80.56 | 18.70 | 9.15 | 7.47 | 78.27 |

Note: All models have been pretrained on Kinetics400 dataset Kay et al. (2017) and continually trained on MicroG-4M. TC denotes the temporal configuration (frame length × sampling rate). #Params indicates the number of parameters (in millions, M).

Table 2: Zero-shot performance on MicroG-4M test set for models pretrained on Kinetics and fine-tuned on AVA Gu et al. (2018).

| Model | | | | | Test Result | | | |
|---|---|---|---|---|---|---|---|---|
| Arch | TC | Backbone | Pretrain | Fine-tune | mAP (%) | F1-score (%) | Recall (%) | AUROC (%) |
| Slow Feichtenhofer et al. (2019) | 4x16 | R50 He et al. (2016) | Kinetics 400 Kay et al. (2017) | AVA v2.2 Gu et al. (2018) | 15.74 | 4.15 | 3.35 | 72.62 |
| Slow Feichtenhofer et al. (2019) | 8x8 | R50 He et al. (2016) | Kinetics 400 Kay et al. (2017) | AVA v2.2 Gu et al. (2018) | 16.24 | 2.67 | 1.99 | 73.83 |
| Slow Feichtenhofer et al. (2019) | 32x2 | R50 He et al. (2016) | Kinetics 400 Kay et al. (2017) | AVA v2.2 Gu et al. (2018) | 11.02 | 0.87 | 0.58 | 67.26 |
| SlowFast Feichtenhofer et al. (2019) | 8x8 | R101 He et al. (2016) | Kinetics 600 Carreira et al. (2018) | AVA v2.2 Gu et al. (2018) | 14.95 | 1.99 | 1.35 | 71.01 |
| SlowFast Feichtenhofer et al. (2019) | 32x2 | R101 He et al. (2016) | Kinetics 600 Carreira et al. (2018) | AVA v2.2 Gu et al. (2018) | 23.81 | 6.32 | 6.62 | 77.83 |
| SlowFast Feichtenhofer et al. (2019) | 16x8 | R101 He et al. (2016) | Kinetics 600 Carreira et al. (2018) | AVA v2.2 Gu et al. (2018) | 15.08 | 0.70 | 0.42 | 69.14 |

Note: All metrics are macro-averaged over action classes. mAP is measured at IoU = 0.5. F1 and AUROC are computed per class and then averaged. TC denotes the temporal configuration (frame length × sampling rate).

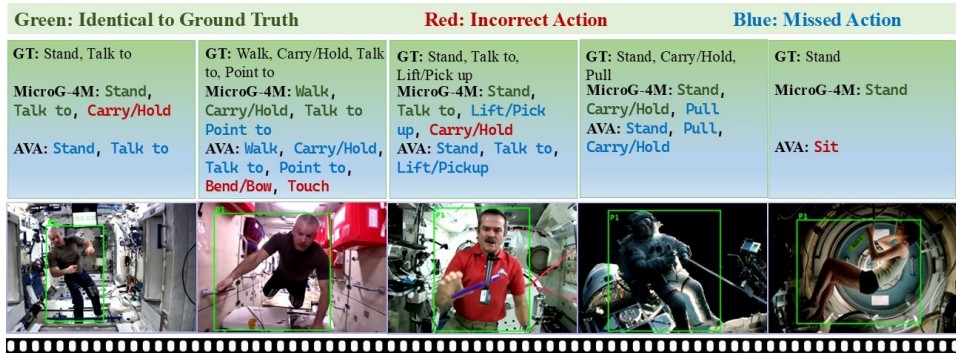

Figure 4: Qualitative results for fine-grained action recognition in microgravity. GT denotes ground truth; MicroG-4M and AVA refer to models fine-tuned on MicroG-4M and AVA, respectively. The MicroG-4M model yields more accurate predictions.

gap beyond conventional terrestrial shifts. Comparing to AVA-finetuned models (Table 2), we observe a sharp drop when transferring directly from AVA Gu et al. (2018) to MicroG-4M despite identical pretraining/backbones; e.g., SlowFast 32×2 reaches only 23.81% mAP on MicroG-4M, far below its MicroG-tuned counterpart. This indicates that Earth-trained assumptions about orientation, support/contact, and object dynamics are fragile in microgravity and are not resolved by naïve fine-tuning alone. Together, these findings position MicroG-4M as a diagnostic benchmark that surfaces gravity-dependent failure modes and motivates methods for robust, space-adapted video understanding. Our dataset thus provides a rigorous testbed for evaluating and advancing space-adapted video understanding models, especially in the context of astronaut assistance and autonomous system development.

**Qualitative Analysis.** Figure 4 presents qualitative comparisons between models trained on AVA Gu et al. (2018) and those fine-tuned on MicroG-4M, across 5 representative video clips captured inside and outside spacecraft cabins. The MicroG-4M–trained model demonstrates high alignment with ground truth labels for core actions such as "Stand", "Walk", and "Talk-to", while the AVA Gu et al. (2018)-based counterpart consistently misinterprets floating or inverted postures as "Bend/Bow" or "Sit", revealing its reliance on Earth-centric gravitational priors. A representative example in the fifth

Table 3: Cross-domain transfer under matched AVA fine-tuning (zero-shot evaluation).

| Model | TC | Backbone | Test Set | mAP (%) | AUROC (%) |
|---|---|---|---|---|---|
| SlowFast Feichtenhofer et al. (2019) | 32×2 | R101 He et al. (2016) | JHMDB Jhuang et al. (2013) | 47.50 | 83.98 |
| SlowFast Feichtenhofer et al. (2019) | 32×2 | R101 He et al. (2016) | MicroG-4M | 23.81 | 77.83 |
| Slow Feichtenhofer et al. (2019) | 8×8 | R50 He et al. (2016) | JHMDB Jhuang et al. (2013) | 34.24 | 76.96 |
| Slow Feichtenhofer et al. (2019) | 8×8 | R50 He et al. (2016) | MicroG-4M | 16.24 | 73.83 |

Table 4: Comparison of video captioning performance across open-source and closed-source models on the MicroG-4M benchmark. "#f" denotes the number of input frames used during model inference.

| Model | #f | CIDEr | BLEU-4 | Rouge-L | Meteor | S-BERT | BERTScore |
|---|---|---|---|---|---|---|---|
| Open-Source | | | | | | | |
| Video-ChatGPT Maaz et al. (2024) | 3 | 0.06 | 0.12 | 10.10 | 4.33 | 39.61 | 85.40 |
| mPLUG-Owl3 Ye et al. (2025) | 3 | 0.16 | 0.40 | 11.87 | 5.88 | 47.45 | 85.91 |
| LLaVA-NeXT Zhang et al. (2024) | 8 | 0.30 | 1.88 | 16.32 | 14.45 | 54.16 | 84.98 |
| Video-LLaVA Lin et al. (2024) | 8 | 0.03 | 0.07 | 9.29 | 4.12 | 42.97 | 84.89 |
| Qwen2.5-VL Yang et al. (2024) | 9 | 0.03 | 1.34 | 13.75 | 15.67 | 56.46 | 84.01 |
| Tarsier2-Recap-7B Yuan et al. (2025) | 16 | 0.04 | 0.03 | 0.17 | 0.12 | 51.35 | 84.53 |
| InternVideo Wang et al. (2022) | 90 | 0.77 | 2.60 | 16.57 | 15.18 | 55.28 | 85.41 |
| Closed-Source | | | | | | | |
| GPT-4o Hurst et al. (2024) | 6 | 1.74 | 2.65 | 16.46 | 11.27 | 62.18 | 86.75 |
| Gemini 1.5 Pro Reid et al. (2024) | 16 | 3.52 | 3.28 | 17.34 | 15.19 | 63.38 | 86.25 |
| Gemini 2.5 Pro Comanici et al. (2025) | 3 | 3.27 | 3.10 | 17.92 | 13.08 | 62.82 | 86.95 |

column shows the AVA Gu et al. (2018)-finetuned model misclassifying a floating astronaut as "Sit" while the MicroG-4M–trained model correctly predicts "Stand" reflecting a common correction of gravity-induced biases. Models trained on MicroG-4M also demonstrate improved robustness in distinguishing passive object drift from intentional manipulation, though semantic ambiguity remains, *e.g.*, predicting "Carry/Hold" when a tool drifts near the astronaut's hand without actual interaction. These results indicate that MicroG-4M mitigates terrestrial biases, enhances sensitivity to body-object dynamics, and better captures domain-specific actions, supporting future work on temporal coherence and intentionality modeling.

## 5.3 EVALUATION OF VIDEO CAPTIONING MODELS

The results in Table 4 reveal how video-language models perform on the MicroG-4M benchmark, highlighting key challenges introduced by microgravity-specific content. Lexical metrics such as CIDEr and BLEU-4 show low overall values, especially among open-source models, suggesting a significant distributional shift between MicroG-4M and typical pretraining data. The dataset's domain-specific vocabulary, visually compositional scenes, and semantically dense annotations likely reduce surface-level overlap, which these metrics are sensitive to. In contrast, semantic similarity metrics such as S-BERT and BERTScore remain relatively higher and more consistent, indicating that several models capture the underlying intent even without lexical alignment. This underscores the semantic richness of MicroG-4M, where alternative phrasings and scientific terminology often convey similar meanings. Performance differences further reveal that input frame density and pretraining modality play key roles. InternVideo Wang et al. (2022), which processes 90 frames sampled within a 3s window, consistently outperforms other open-source models. This suggests that dense sampling, coupled with video-specific pretraining, enhances the model's ability to capture subtle spatial patterns and object-scene relationships—features that are particularly important in microgravity scenarios, where visual cues are often atypical or physically ambiguous. Closed-source models, *i.e.*, GPT-4o Hurst et al. (2024), Gemini 1.5 Pro Reid et al. (2024) and Gemini 2.5 Pro Comanici et al. (2025) achieve better scores on both lexical and semantic metrics, likely due to broader data exposure, larger capacity, or more advanced cross-modal fusion strategies. However, their relatively small performance gains further validate the challenge posed by MicroG-4M. In general, these findings position MicroG-4M as a demanding benchmark for evaluating multimodal models under domain change, highlighting the need for robust spatial reasoning, domain adaptation, and semantically aware generation strategies in unconventional environments.

Table 5: Experiments of Visual Question Answering (VQA) models on the MicroG-4M benchmark.

| Model | #f | CIDEr | BLEU-4 | Rouge-L | Meteor | S-VQA | BERTScore |
|---|---|---|---|---|---|---|---|
| Open-Source | | | | | | | |
| LLaVA-NeXT Zhang et al. (2024) | 8 | 24.00 | 22.14 | 15.56 | 12.40 | 38.08 | 87.15 |
| Video-LLaVA Lin et al. (2024) | 8 | 25.70 | 28.47 | 15.90 | 10.71 | 35.39 | 87.13 |
| Qwen2.5-VL Yang et al. (2024) | 9 | 2.99 | 0.65 | 8.35 | 8.47 | 40.65 | 84.80 |
| Tarsier2-Recap-7B Yuan et al. (2025) | 16 | 5.08 | 0.01 | 0.08 | 0.09 | 29.60 | 85.30 |
| Closed-Source | | | | | | | |
| Gemini 1.5 Pro Reid et al. (2024) | 16 | 8.78 | 1.33 | 13.03 | 12.54 | 43.15 | 86.41 |
| Gemini 2.5 Pro Comanici et al. (2025) | 3 | 0.67 | 0.48 | 7.58 | 9.12 | 44.75 | 84.89 |
| GPT-4o Hurst et al. (2024) | 6 | 33.98 | 3.76 | 18.11 | 15.89 | 44.56 | 87.81 |

## 5.4 Evaluation of Visual Question Answering Models

The results in Table 5 demonstrate that MicroG-4M presents distinct challenges for visual question answering. Notably, there is a significant divergence between lexical and semantic evaluation metrics, visible in both open- and closed-source models. For example, Qwen2.5-VL Bai et al. (2025) yields a BLEU-4 of only 0.65 and a CIDEr score of 2.99, yet achieves the highest S-VQA score in its category (40.65). A similar pattern appears for Gemini 2.5 Pro Comanici et al. (2025), which attains the highest S-VQA score among closed-source models (44.75) while exhibiting very low CIDEr and BLEU-4 scores, largely because it tends to answer simple questions with long, explanatory responses. This contrast suggests that MicroG-4M questions often admit multiple semantically valid answers that differ lexically, such as paraphrased actions, scientific terms, or object references adapted to microgravity settings. This characteristic does not reflect inconsistency in evaluation, but rather underscores the conceptual and linguistic diversity embedded in the dataset. In addition, the moderate absolute scores of even the top-performing closed-source models, such as GPT-4o Hurst et al. (2024) (CIDEr 33.98, S-VQA 44.56), reveal the difficulty of reasoning over visual content in microgravity. Unlike conventional VQA datasets, MicroG-4M includes visually ambiguous cues, *e.g.*, floating objects, unusual body orientations, and tool manipulations under microgravity that challenge models trained primarily on terrestrial data. This suggests that current pretraining corpora lack sufficient coverage of such scenarios, and that purely scaling model capacity is insufficient for reliable generalization. Interestingly, increasing the number of input frames within the fixed 3s window does not consistently yield better performance. For example, Gemini 1.5 Pro Reid et al. (2024) processes 16 frames but performs worse than GPT-4o Hurst et al. (2024), which uses only 6 frames, and Gemini 2.5 Pro Comanici et al. (2025) achieves the best S-VQA score with just 3 frames. This indicates that dense frame sampling alone is insufficient. Instead, performance depends more critically on the model's ability to extract semantically salient cues, *e.g.*, astronaut posture, object manipulation, and spatial configurations, from visually subtle or low-motion segments. In microgravity environments, where conventional motion dynamics and object affordances are altered, effective spatial reasoning and cross-modal alignment appear to be more decisive than temporal redundancy. In summary, MicroG-4M reveals key limitations of current VQA systems in addressing domain-specific challenges, particularly those involving spatial complexity, ambiguous motion, and semantically flexible queries inherent to microgravity. Its comprehensive and specialized design establishes it as a valuable testbed for probing the robustness, adaptability, and generalization capabilities of multimodal models well beyond the scope of conventional Earth-based benchmarks.

## 6 Conclusion

In this work, we present MicroG-4M, the first large-scale dataset specifically curated for human action recognition and vision-language understanding in microgravity environments. The dataset features 4, 759 annotated video clips with over 390, 000 bounding boxes and 13, 000+ action labels across 50 unique action classes. It also includes human-written captions and over 7, 000 VQA pairs, enabling rich semantic understanding and reasoning. We introduce MicroG-Bench, a benchmark for evaluating state-of-the-art models in fine-grained action recognition, video captioning, and question answering. Results show significant performance degradation in space-like settings, highlighting the need for domain-specific benchmarks and adaptation. MicroG-4M advances robust, generalizable AI for astronaut support and autonomous space operations.

## ACKNOWLEDGMENTS

The project is funded by the Deutsche Forschungsgemeinschaft (DFG, German Research Foundation) – SFB-1574 – 471687386. This work was supported in part by the SmartAge project sponsored by the Carl Zeiss Stiftung (P2019-01-003; 2021-2026). This work is also supported in part by the National Natural Science Foundation of China under Grant No. 62473139, in part by the Hunan Provincial Research and Development Project (Grant No. 2025QK3019), and in part by the State Key Laboratory of Autonomous Intelligent Unmanned Systems (the opening project number ZZKF2025-2-10). This research was partially funded by the Ministry of Education and Science of Bulgaria (support for INSAIT, part of the Bulgarian National Roadmap for Research Infrastructure).

## ETHICS STATEMENT

This work constructs a benchmark from publicly available videos released by official space agencies and publicly distributed films. No private recordings, surveillance footage, or restricted-access materials are included. For public release, we distribute annotations and metadata only, and require users to obtain source videos independently through legal channels. This avoids redistribution of copyrighted audiovisual content and supports compliance with institutional and regional regulations.

The dataset depicts professional activities of publicly visible astronauts and actors in occupational settings. While publicly disseminated material typically does not require additional individual consent for research use, downstream users are instructed to follow their institutions' IRB/ethics guidelines and applicable privacy regulations. No personally identifying information beyond what is inherently visible in the source videos is collected or released, and metadata exclude personal identifiers.

The dataset does not include satellite imagery or sensitive geolocation data. Film clips are screened for physical plausibility and safety consistency, and implausible scenes are removed. For vision–language components, large language models are used to generate and rank candidate questions under interrogative diversity, with human verification of final questions and answers. Unanswerable "Not mentioned" cases are retained to discourage hallucination and overconfident speculation.

To mitigate misuse, the resource is released for research purposes under terms that prohibit re-identification attempts, surveillance applications, or violations of platform policies and content owners' rights. We will honor takedown requests from rightsholders and provide a mechanism to remove or amend entries if concerns arise. Overall, the release policy combines metadata-only distribution and clear usage terms, which aims to balance openness and reproducibility with privacy, safety, and copyright considerations.

## REPRODUCIBILITY STATEMENT

All data annotations, metadata, and code are publicly available at https://github.com/lei-qi-233/MicroG-4M. We provide scripts for data preprocessing, benchmark construction, and evaluation, along with documentation for reproducing the reported results.

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

APPENDIX

## A  THE USE OF LARGE LANGUAGE MODELS (LLMs)

We use LLMs in two limited ways. During dataset construction, an LLM proposes candidate VQA questions per 3 s clip; humans filter, verify, and finalize both questions and answers, and captions are human written. For benchmarking, several LLM-based systems are evaluated as baselines under a unified inference setup. LLMs do not assign action labels or alter evaluation protocols. We also used an LLM for language polishing of the manuscript, and all technical content and conclusions were written and verified by the authors.

## B  TECHNICAL LIMITATIONS

MicroG-4M introduces the first unified benchmark for high-level video understanding in microgravity environments, but several technical limitations remain that could guide future refinement and expansion.

One challenge lies in the ambiguity and subjectivity of annotations. Interpreting actions and generating captions can vary between annotators, especially in visually ambiguous frames or when inference is required. Even with cross-validation protocols in place, some degree of subjectivity may persist, introducing annotation noise.

Another limitation is the restricted temporal context. Each video clip spans only three seconds, which may hinder the ability to model long-term dependencies. This limitation is particularly relevant when modeling multi-step operations that are common in space missions.

The dataset is currently limited to RGB visual input, which constrains the potential for multimodal understanding. Future versions could benefit from the inclusion of additional modalities such as audio signals or communication transcripts to support more comprehensive reasoning.

There is also an inherent domain bias introduced by the inclusion of cinematic footage. While these clips are visually high-fidelity and physically plausible, they may differ in visual style and narrative framing from real operational recordings. This discrepancy can affect the generalizability of models trained on the dataset.

Finally, the scale and annotation coverage of the dataset present further constraints. The current release offers a carefully annotated subset of the full video collection, suitable for benchmarking, but smaller than many large-scale web datasets. Continued annotation efforts are underway to expand the dataset, covering a wider range of clips, actions, and scene types to support more diverse downstream applications, including temporal reasoning and sequence-level inference.

## C  POTENTIAL SOCIAL IMPACTS

MicroG-4M introduces the first benchmark specifically designed for video understanding and vision-language reasoning in microgravity environments. It comprises 4,759 curated clips supporting fine-grained action recognition, video captioning, and visual question answering. These tasks collectively provide a testbed for evaluating models under the unique motion dynamics and spatial ambiguities posed by microgravity.

While MicroG-4M is not intended for deployment, it may inform downstream research in areas such as astronaut behavior modeling, procedural understanding, or human–robot collaboration. Its vision-language annotations also facilitate studies on video summarization and temporal grounding, with potential implications for future human-AI interfaces in space-based or analog settings.

MicroG-4M enables the analysis of model limitations under microgravity-specific challenges, such as sensitivity to gravitational priors and orientation ambiguity, offering a foundation for research on robustness and generalization. Although developed for space-based contexts, the dataset's motion and interaction patterns may inspire comparative studies in gravity-reduced analog environments, such as underwater settings.

Furthermore, the semantic complexity of the captioning and QA tasks highlights challenges including hallucination and semantic inconsistency, positioning MicroG-4M as a testbed for evaluating the reliability and grounding of multimodal models in physically unfamiliar conditions.

## D SAFETY AND ETHICAL DISCUSSION

MicroG-4M is developed as a research-oriented benchmark for video understanding in microgravity environments. While ethical standards were followed throughout its construction, several considerations are noted to promote responsible and informed use.

All real-world videos are sourced from publicly available materials released by official space agencies and educational institutions, containing no private or sensitive content. Astronauts are shown exclusively in professional contexts. Simulated cinematic content, while enriching visual diversity, may introduce stylistic bias that differs from real operational footage.

All captions and QA annotations were created manually by annotators with domain guidance. LLMs were employed exclusively for grammatical correction and fluency enhancement, without contributing to semantic generation. Nonetheless, users should remain aware of any residual stylistic biases introduced during this refinement process.

MicroG-4M is intended solely for non-commercial academic research. It is not validated for real-world deployment, particularly in sensitive domains such as surveillance or defense. Users are advised to evaluate generalization carefully and avoid overextension of model outputs.

We encourage community feedback on potential biases, content issues, or safety risks. Future versions will include expanded validation and filtering to enhance transparency and data quality.

## E ADDITIONAL QUALITATIVE ANALYSIS

### E.1 FINE-GRAINED HUMAN ACTION RECOGNITION

We provide qualitative comparisons using representative video samples to illustrate the fine-grained action recognition performance of different models in microgravity scenarios (Figure 5). These examples reveal two consistent signatures that drive the domain gap between Earth-trained and microgravity-tuned models.

First, gravity-dependent person-movement classes are fragile. When the vertical reference is ambiguous or decoupled from posture, the AVA-finetuned model projects microgravity poses onto terrestrial categories such as *bend/bow*, *crouch/kneel*, *sit*, or *stand*. Representative cases include: hatch crossing labeled as *bend/bow* (upper panel, column 1); free-floating or foot-restrained standing either mislabeled as *sit* (upper panel, column 3; lower panel, column 1) or not detected at all (upper panel, column 4; lower panel, column 5); sideways *crouch/kneel* missed entirely (lower panel, last column); and microgravity *sit* mapped to *lie/sleep* due to orientation mismatch (upper panel, column 5). These observations indicate that person-movement classes that implicitly assume an upright axis are unstable in orbit.

Second, transient force direction and short-contact object manipulation are difficult. Short episodes of *lift/pick up* and *put down* are detected inconsistently or absorbed into background motion (upper panel, column 2; lower panel, column 2). The action *hit (an object)* receives low confidence and is suppressed by the detection threshold, yielding no action label (lower panel, rightmost column). In contrast, device-centric actions with clear visual anchors and sustained contact, most notably *carry/hold (an object)*, are comparatively stable and predicted consistently across models in the same examples (upper panel, column 2; lower panel, columns 2–4).

### E.2 VIDEO CAPTIONING

We provide qualitative examples to emphasize the unique challenges posed by space-related scenarios and the limitations of existing state-of-the-art multimodal models (mPLUG-Owl3, LLaVA-NeXT, GPT-4o, Gemini 1.5 Pro) in accurately capturing specialized details inherent to space station environments.

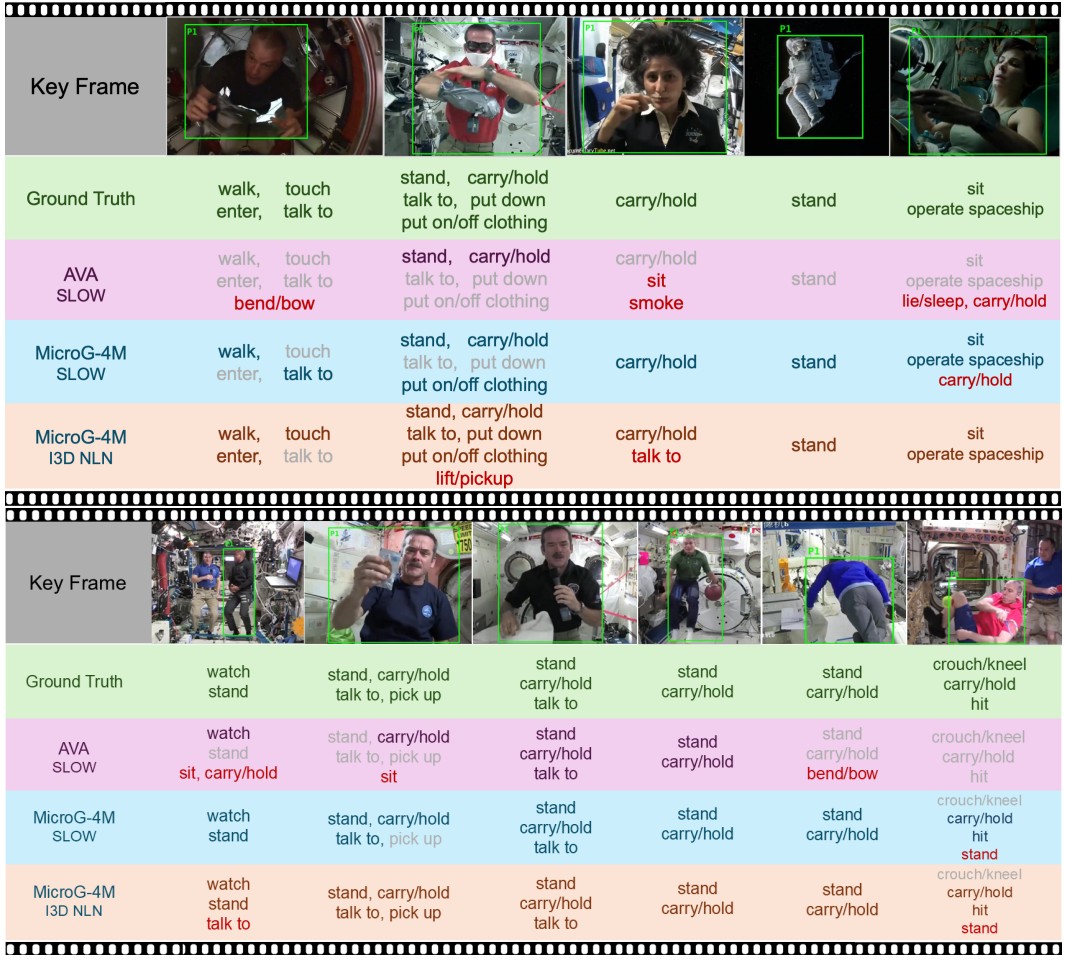

Figure 5: Qualitative results for fine-grained human action recognition in microgravity. Below the frame samples, the first row presents the ground truth labels of the actions. The second row presents the predictions of the Slow architecture fine-tuned on the AVA dataset. The third row shows the predictions of the Slow architecture fine-tuned on the MicroG-4M dataset. The last row shows the predictions of the I3D Non-Local Network (NLN) architecture fine-tuned on MicroG-4M. For both the Slow and I3D NLN architectures, the AVA and MicroG-4M models were trained under the same configuration: a ResNet-50 backbone with an 8×8 input (frame length × sampling rate), pre-trained on Kinetics-400. The I3D NLN model fine-tuned on MicroG-4M achieved the highest mAP among our baselines. Gray text denotes missed detections, while red text denotes false detections.

In the first scenario (Figure 6, left), all models demonstrate significant shortcomings in capturing critical, astronaut-specific information and precise operational context, underscoring the difficulty in accurately describing space-station activities without access to specialized annotations.

The second scenario (Figure 6, right) represents a relatively simpler context, yet models still lack precise details and fail to fully leverage the specialized information present in our ground-truth annotations. These examples illustrate the inherent difficulty in accurately modeling highly specialized and contextually rich scenarios typical of space environments, highlighting the necessity and distinctive value of our carefully annotated space-oriented dataset.

Overall, these qualitative results demonstrate the critical importance of domain-specific annotation for effectively capturing the nuanced details and specialized context essential in space exploration scenarios.

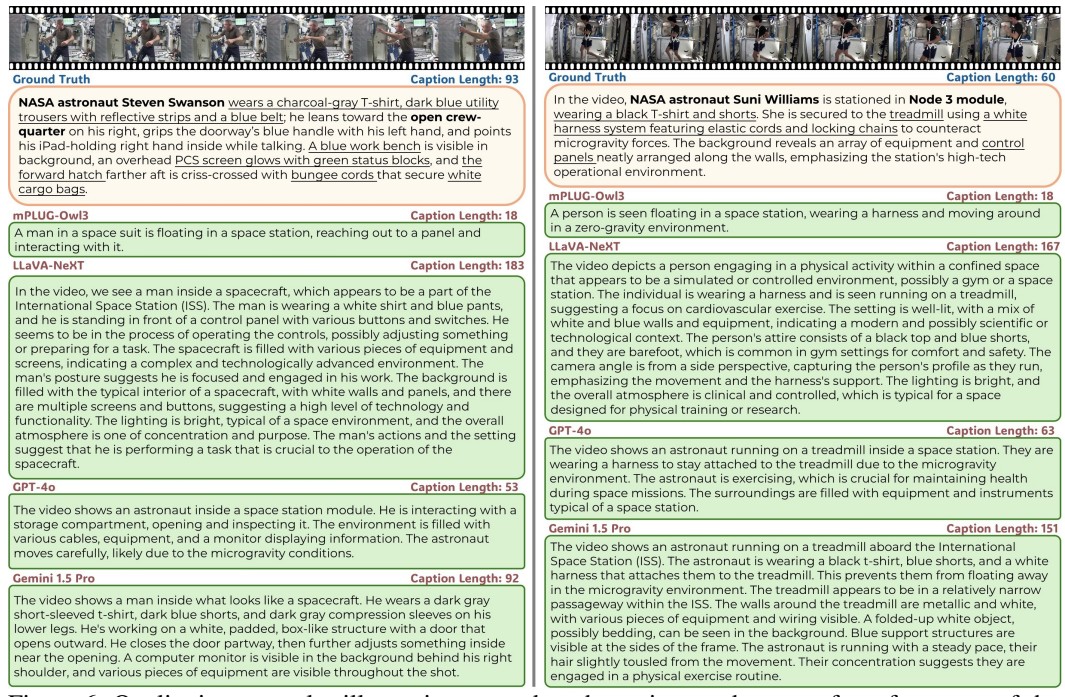

Figure 6: Qualitative examples illustrating ground-truth captions and outputs from four state-of-the-art multimodal models. The left example represents a challenging scenario, in which all models fail to accurately capture detailed and precise information. The right example demonstrates a relatively simpler scenario, where model-generated captions exhibit closer alignment to the ground truth. Each caption includes the corresponding caption length (in words), with key details highlighted in the ground-truth captions.

### E.3 VIDEO QUESTION ANSWERING

We present qualitative examples illustrating the performance of state-of-the-art multimodal models (Gemini 1.5 Pro, GPT-4o, Video-LLaVA) on challenging Video Question-Answering (VQA) tasks specifically related to space environments.

The provided examples (Figure 7) showcase diverse and challenging scenarios uniquely associated with space-based contexts. The top-left example demonstrates the complexity of accurately interpreting multiple astronauts' actions within confined spaces, especially when occlusions occur. The bottom-left example emphasizes challenges in accurately identifying specific locations unique to space environments. The top-right scenario tests models' abilities to describe various specialized objects within densely detailed backgrounds, typical of spacecraft interiors. Lastly, the bottom-right example addresses the models' propensity for hallucination by evaluating their capacity to accurately infer details not explicitly mentioned in the visual content.

Overall, these qualitative analyses highlight significant limitations of current multimodal models in handling space-specific scenarios, underscoring the critical need for detailed and specialized annotations provided by our dataset to enhance performance and robustness in space-related VQA tasks.

## F DATA FORMAT CONVERSION

To simplify frame-level annotation processing, we adapted our dataset to match the AVA format, with one modification: using frame stamps rather than timestamps. This allows direct frame indexing, aligning better with our annotation scheme and reducing unnecessary conversion overhead.

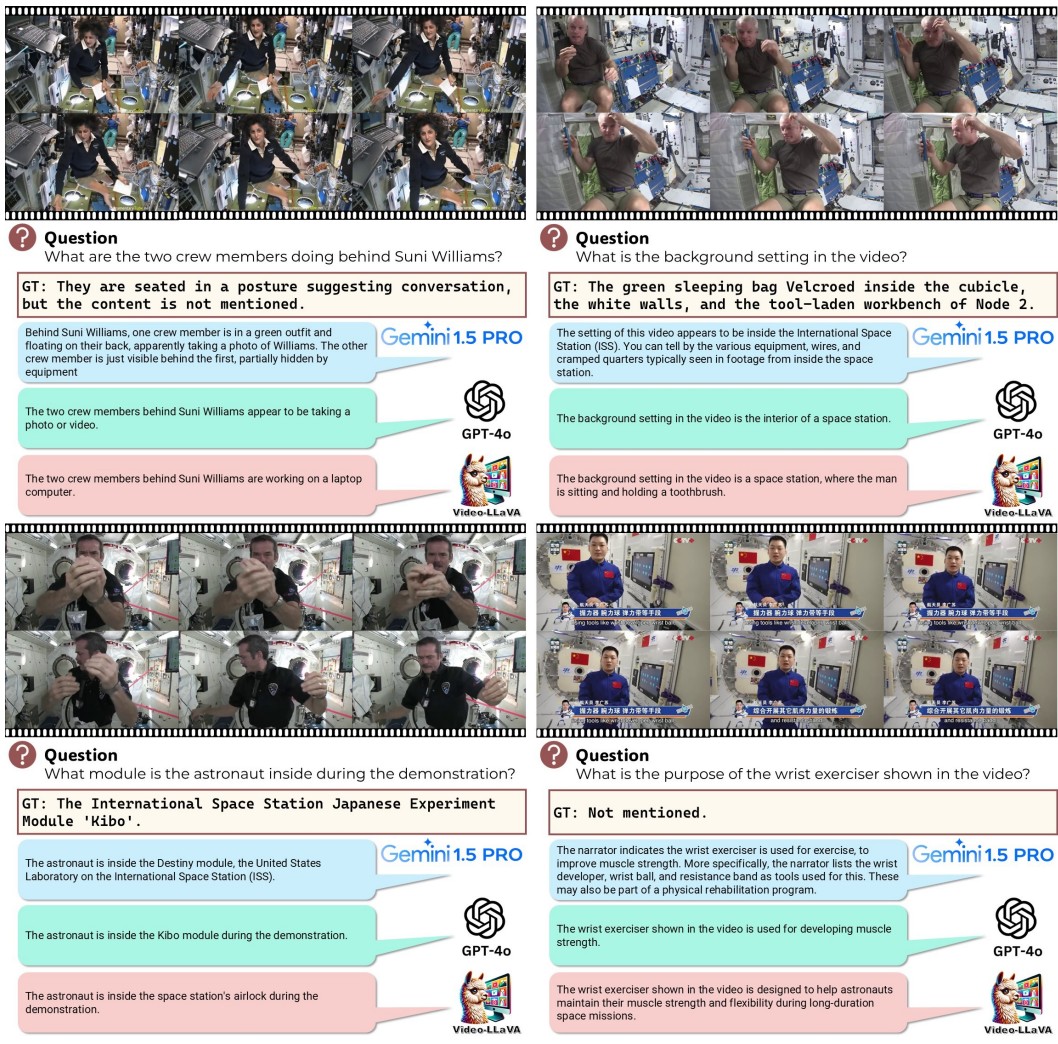

Figure 7: Qualitative examples illustrating Ground-Truth (GT) answers and corresponding responses generated by three state-of-the-art multimodal models.

## F.1 DATASET METADATA

Our dataset metadata is structured hierarchically into two categories—video-level and frame-level—to facilitate efficient retrieval and flexible querying.

**Video-Level Metadata:** summarizes high-level annotations and semantic context:

- **Video ID**: Unique identifier for each video clip.
- **Person ID**: Unique identifier for annotated individuals within videos.
- **Action Labels**: Fine-grained human action labels.
- **Caption**: Detailed textual description of the video content.
- **Question-Answer Pairs**: Structured questions and answers related to video content.

**Frame-Level Metadata:** provides detailed annotations at individual frame granularity:

- **Video ID**: Corresponding identifier linking video-level metadata.
- **Frame stamp**: Temporal location of annotated frames within videos.
- **Person ID**: Unique identifier for each individual per frame.

- **Bounding Box**: Pixel coordinates (xmin, ymin, xmax, ymax) for each annotated person.
- **Clip Type**: Indicates real-world or cinematic origin of video clips.

Video naming follows the "[source identifier]_[sequence number]" format, where the identifier is the YouTube ID or film name, and the sequence number indicates its temporal position. This structured metadata approach ensures efficient integration between frame-level and video-level annotations, effectively supporting downstream vision-language research in microgravity environments. For comprehensive documentation of the dataset metadata, please refer to the resources provided on our GitHub and Hugging Face repositories.

# G    Supplementary for Human Action Recognition (HAR) Task

## G.1    Dataset Partitioning

The dataset was partitioned at the video level following two primary criteria: (1) ensuring coverage of all action classes through greedy selection, and (2) proportional random splitting of remaining videos into training, validation, and test subsets (70:10:20 ratio). This approach avoids label leakage by maintaining video-level annotation consistency within subsets. Table 6 provides a summary of the sample-level and video-level distributions across splits.

Table 6: Train/val/test split statistics for the HAR task in MicroG-4M. Each cell reports both the absolute count and its corresponding percentage (%).

| Split | Samples (# / %) | Video Clips (# / %) |
|-------|-----------------|---------------------|
| Train | 9,273 (69.93%) | 3,331 (69.99%) |
| Val | 1,330 (10.03%) | 475 (9.98%) |
| Test | 2,658 (20.04%) | 953 (20.03%) |
| Total | 13,261 (100.00%) | 4,759 (100.00%) |

## G.2    Dataset Statistics

Table 7 summarizes the distribution of three broad action types, namely Object Manipulation, Person Interaction, and Person Movement, across the entire dataset, cinematic subset, and real video subset. Object Manipulation constitutes the largest category, particularly within real videos (39.42%), followed by Person Interaction and Person Movement categories.

Table 8 presents statistics regarding the number of persons annotated per video. Single-person videos dominate the dataset, particularly in real footage (92.41%), while multi-person annotations are comparatively more frequent in cinematic sources.

Additionally, Table 9 provides a detailed breakdown of 50 fine-grained action classes. The most frequent actions across both subsets include 'stand', 'carry/hold object', and 'talk to self/person', whereas actions such as 'climb', 'take photo', and 'kiss person' are comparatively rare. Notably, the distribution of action labels exhibits a pronounced long-tail pattern, consistent with Zipf's law commonly observed in naturally occurring datasets, indicating the realistic and representative nature of the collected action annotations.

Table 7: Label type distribution across the full dataset, cinematic subset, and real video subset. Each cell shows the number of labels followed by its proportion (%).

| Label Type | All Videos | Movies | Real Videos |
|------------|------------|--------|-------------|
| Object Manipulation | 4,986 (37.60%) | 1,416 (38.78%) | 3,788 (39.42%) |
| Person Interaction | 4,288 (32.34%) | 1,198 (32.81%) | 2,950 (30.70%) |
| Person Movement | 3,987 (30.07%) | 1,037 (28.40%) | 2,872 (29.89%) |

Table 8: Distribution of the number of persons per video in MicroG-4M. Each cell shows the number of videos followed by its proportion (%).

| Persons per Video | All Videos | Movies | Real Videos |
|---|---|---|---|
| Single Person (1) | 3,983 (83.69%) | 816 (61.26%) | 3,167 (92.41%) |
| Two Persons (2) | 623 (13.09%) | 395 (29.65%) | 228 (6.65%) |
| Three or More ($\geq$3) | 153 (3.22%) | 121 (9.09%) | 28 (0.94%) |

### G.3 PER-CLASS AP RESULTS

Building on the aggregate results in Table 1 and Table 2, we analyze per-class AP on the MicroG-4M test set. Figures 8–14 report the per-class curves with the 50 AVA-derived MicroG-4M actions ordered as in Table 9.

**Paired comparison.** For **Slow** and **SlowFast**, we provide matched plots under both training regimes, MicroG-4M→MicroG-4M and AVA→MicroG-4M (Figures 8, 9, 10, 11). MicroG-4M fine-tuning lifts per-class AP on gravity-dependent person-movements and transient-contact object manipulations, including *lie/sleep*, *run/jog*, *sit*, *close (e.g., a door, a box)*, *enter*, *hit (an object)*, *lift/pick up*, *put down*, *work on a computer*, and *hug (a person)*. The AVA→MicroG-4M counterparts drop on the same groups, aligning with the quantitative plateau and ranking inversion discussed earlier and with the qualitative signatures that microgravity removes gravity-aligned posture cues and shortens hand–object contact.

**Across remaining backbones.** For **C2D**, **I3D**, **MViT** family, and **X3D**, MicroG-4M fine-tuning generally improves the posture-sensitive actions and short-contact manipulations above, yet several classes remain hard across architectures: *pull (an object)*, *push (an object)*, *close* (e.g., *a door, a box)*, *take a photo*, *text on/look at a cellphone*, *fight/hit (a person)*, and *take (an object) from (a person)*. These actions share low visual salience or momentary cues such as small handheld objects, brief hand–hand transfers, global camera/scene drift in spacecraft cabins, which limits per-class AP even after in-domain training (see Figures 12, 13, 14, and 15).

**Backbone-specific observations.** **MViTv1**, **MViTv2**, and **X3D** exhibit comparatively limited gains after MicroG-4M fine-tuning. Two factors are consistent with this trend. First, temporal windowing and sampling may not balance long drift with short contact, which requires both extended context and reliable short-term discrimination. Second, inductive biases differ: backbones with stronger local encoding and non-local augmentation align better with microgravity-specific postures and contact cues than global-attention–dominant designs with weaker local bias or very lightweight CNNs without non-local pathways. These patterns are visible in the per-class plots for X3D and MViT (Figures 14 and 15), and they are consistent with the aggregate results in Tables 1 and 2, which together explain the test mAP plateau and indicate that in-domain fine-tuning recovers part of the gap while several action categories remain challenging in orbit.

### G.4 CROSS-DATASET VQA DENSITY AND DESIGN RATIONALE

To contextualize our choice of six QA pairs per 3-second clip (**2 QA/s**), we compare MicroG-4M with representative video-VQA corpora in terms of clip granularity, QA density, and question types. This unified view allows controlled discussion of annotation budget and reasoning load across datasets with heterogeneous clip lengths.

MicroG-4M targets short clips at high QA density (2.00 QA/s), comparable to MSVD-QA and exceeding other widely used corpora. A fixed per-clip budget supports semantic diversity while avoiding redundancy typical of long-form settings. Our QA taxonomy balances *foreground actions*, *spatial context*, *entity/attribute grounding*, and *temporal/causal reasoning*, and explicitly includes an *unanswerable* option to reduce hallucination.

Because question generation protocols and scoring schemes vary across datasets, cross-corpus density should be interpreted as a *comparability aid* rather than a difficulty metric. The choice of six QA

Table 9: Complete summary of action class distribution across all videos, movies, and real videos (sorted by action ID).

| ID | Action Name | Count | | |
|---|---|---|---|---|
| | | **All Videos** | **Movies** | **Real Videos** |
| 1 | bend/bow (at waist) | 26 | 14 | 12 |
| 3 | crouch/kneel | 20 | 2 | 18 |
| 5 | fall down | 10 | 1 | 9 |
| 6 | get up | 25 | 4 | 21 |
| 7 | jump/leap | 20 | 0 | 20 |
| 8 | lie/sleep | 17 | 4 | 13 |
| 9 | martial art | 18 | 0 | 18 |
| 10 | run/jog | 12 | 0 | 12 |
| 11 | sit | 252 | 239 | 13 |
| 12 | stand | 3218 | 698 | 2520 |
| 14 | walk | 369 | 75 | 294 |
| 17 | carry/hold object | 3126 | 549 | 2577 |
| 20 | climb (e.g., mountain) | 1 | 1 | 0 |
| 22 | close (door/box) | 13 | 4 | 9 |
| 24 | cut | 9 | 0 | 9 |
| 26 | dress/undress clothing | 31 | 9 | 22 |
| 27 | drink | 43 | 5 | 38 |
| 28 | operate spaceship | 20 | 16 | 4 |
| 29 | eat | 45 | 2 | 43 |
| 30 | enter | 68 | 5 | 63 |
| 34 | hit object | 33 | 3 | 30 |
| 36 | lift/pick up | 188 | 10 | 178 |
| 38 | open (window/door) | 32 | 13 | 19 |
| 41 | play musical instrument | 3 | 0 | 3 |
| 43 | point to object | 323 | 2 | 321 |
| 45 | pull object | 32 | 19 | 13 |
| 46 | push object | 24 | 8 | 16 |
| 47 | put down | 138 | 7 | 131 |
| 48 | read | 15 | 14 | 1 |
| 56 | take photo | 2 | 0 | 2 |
| 57 | text/look at cellphone | 7 | 0 | 7 |
| 58 | throw | 4 | 0 | 4 |
| 59 | touch object | 353 | 136 | 217 |
| 60 | turn screwdriver | 17 | 9 | 8 |
| 61 | watch TV/unspecified | 346 | 316 | 30 |
| 62 | work on computer | 110 | 67 | 43 |
| 63 | write | 3 | 3 | 0 |
| 64 | fight/hit person | 27 | 26 | 1 |
| 65 | give/serve object | 46 | 20 | 26 |
| 66 | grab person | 41 | 31 | 10 |
| 67 | hand clap | 3 | 3 | 0 |
| 68 | hand shake | 4 | 2 | 2 |
| 69 | hand wave | 140 | 44 | 96 |
| 70 | hug person | 16 | 13 | 3 |
| 72 | kiss person | 1 | 1 | 0 |
| 74 | listen to person | 148 | 135 | 13 |
| 76 | push person | 3 | 2 | 1 |
| 78 | take object from person | 15 | 10 | 5 |
| 79 | talk to self/person | 3131 | 504 | 2627 |
| 80 | watch person | 713 | 625 | 88 |

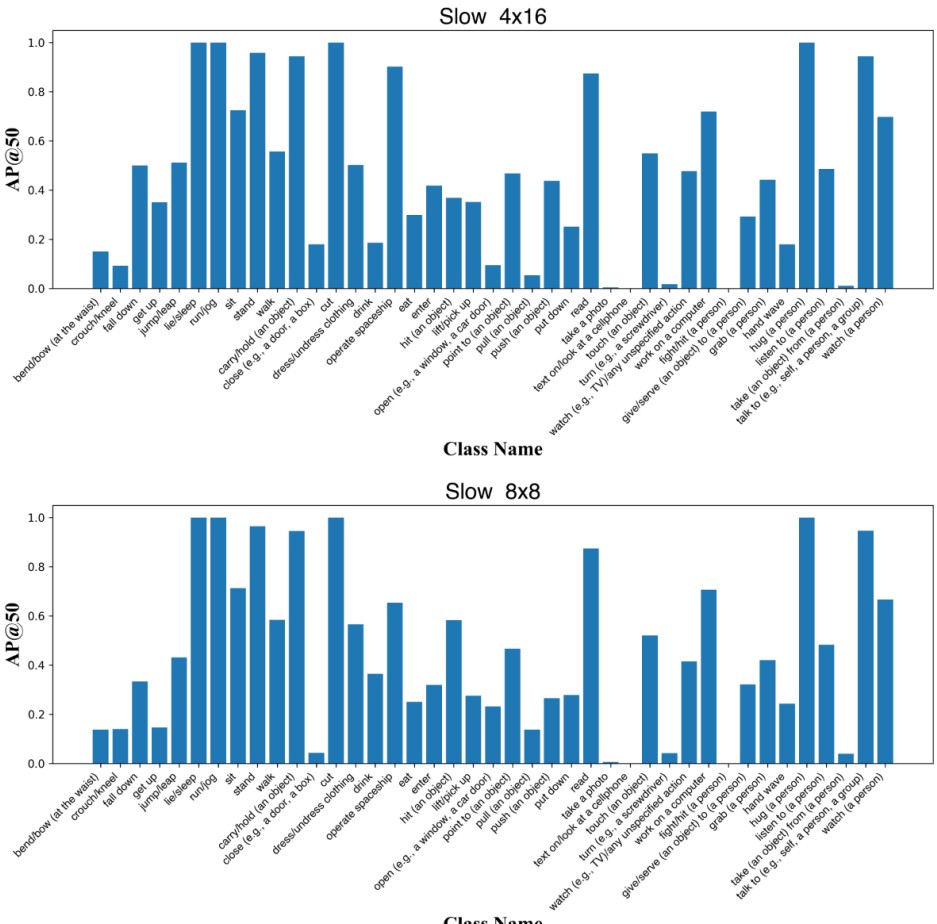

Figure 8: Per-class AP (%) on the MicroG-4M test set for **Slow** Feichtenhofer et al. (2019). Training regime: MicroG-4M fine-tuning. AP@0.5 over the 50 MicroG-4M action classes; class order matches Table 9. Legend uses $L \times S$ to denote clip length $\times$ sampling stride; settings match Table 1.

Table 10: Cross-dataset VQA density and scope. Statistics are compiled from the original dataset reports. "QA/sec" denotes (Avg. QA/Clip)/(Clip length in seconds). This comparison contextualizes our choice of six QA per 3-second clip in MicroG-4M.

| Dataset | Clips | QA pairs | Avg. QA/Clip | Clip Len. (s) | QA/sec | QA Types |
|---|---|---|---|---|---|---|
| MSVD-QA Xu et al. (2017) | 1,970 | 50,505 | 25.64 | 10 | 2.56 | *Wh*-type |
| MSRVTT-QA Xu et al. (2017) | 10,000 | 243,680 | 24.37 | 15 | 1.62 | *Wh*-type |
| TGIF-QA Jang et al. (2017) | 56,720 | 103,919 | 1.83 | 3 | 0.61 | Task-based |
| TVQA Lei et al. (2018) | 21,793 | 152,545 | 7.00 | 76 | 0.09 | *Wh*-type + Temporal |
| ActivityNet-QA Yu et al. (2019) | 5,800 | 58,000 | 10.00 | 180 | 0.06 | Motion, Spatial, Temporal |
| MovieQA Tapaswi et al. (2016) | 6,771 | 6,462 | 0.95 | 200 | 0.004 | Story comprehension |
| VideoQA Yang et al. (2003) | 18,100 | 174,775 | 9.66 | 45 | 0.21 | *Wh*-type + Yes/No |
| **MicroG-4M (Ours)** | **1,238** | **7,428** | **6.00** | **3** | **2.00** | *Wh*-type, Foreground/Background, Fine-/Coarse-motion, Identity, Temporal, Causal |

pairs per 3-second segment is thus motivated by controllability: it preserves clip-level coverage and evaluation stability while enabling *apples-to-apples* studies of domain shift in microgravity, independent of confounds from variable clip durations or QA volumes.

## G.5 EVALUATION WITH GEMINI 2.5 PRO ON THE HAR TASK

We additionally evaluate a strong multimodal large language model, Gemini 2.5 Pro Comanici et al. (2025), on the MicroG-4M human action recognition (HAR) task. We restrict the label space to the

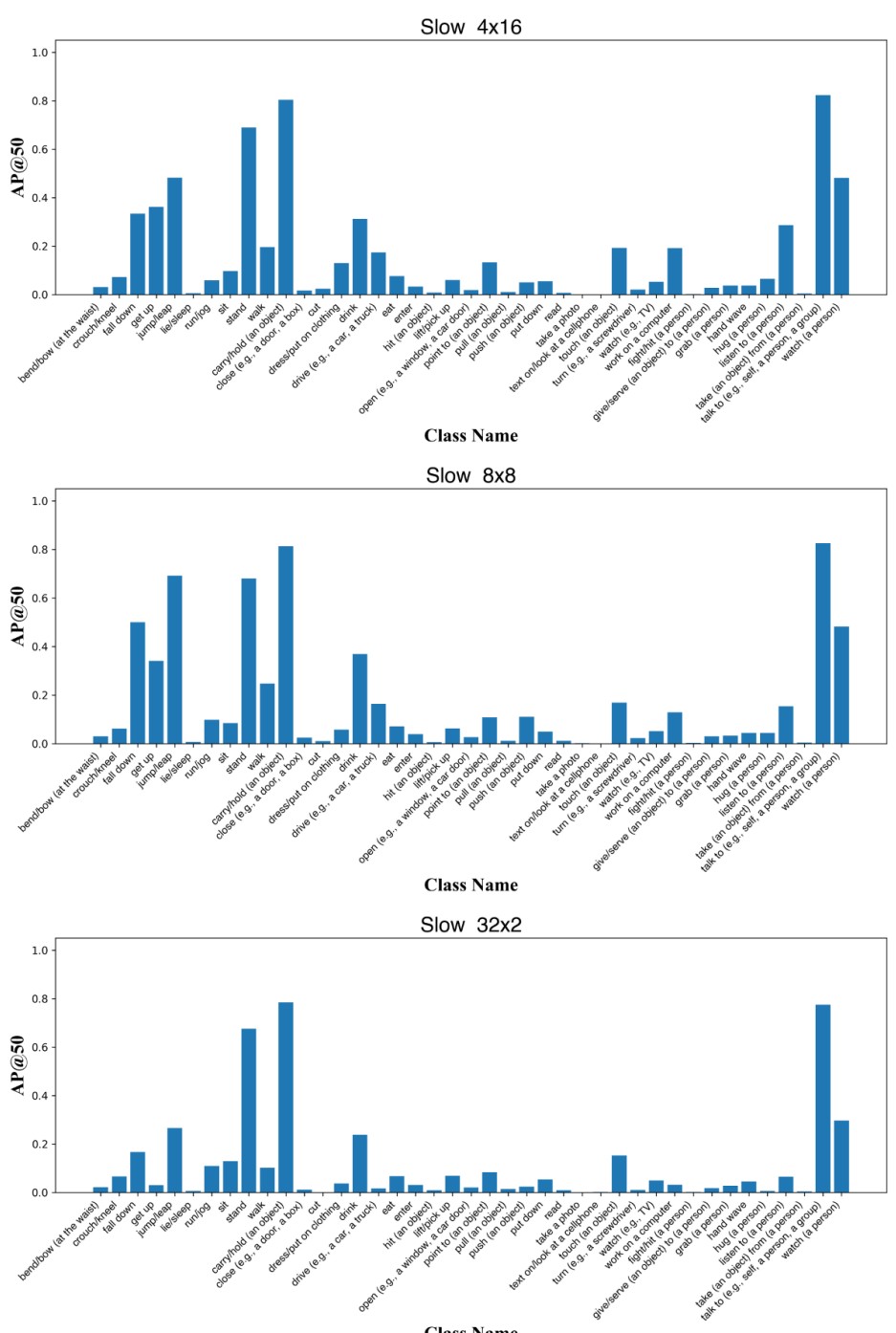

Figure 9: Per-class AP (%) on the MicroG-4M test set for **Slow** Feichtenhofer et al. (2019). Training regime: AVA fine-tuning with zero-shot evaluation on MicroG-4M. AP@0.5 over the 50 AVA-derived action classes. Legend uses $L \times S$ to denote clip length $\times$ sampling stride; settings match Table 1.

50 AVA-derived action classes used in our HAR experiments and formulate HAR as a multi-label prediction problem over these 50 classes. For each 3-second clip and tracked person, Gemini is provided with a per-person video in which the target individual is highlighted by a red bounding box, together with a prompt listing all 50 action IDs and names. The model is instructed to return a list

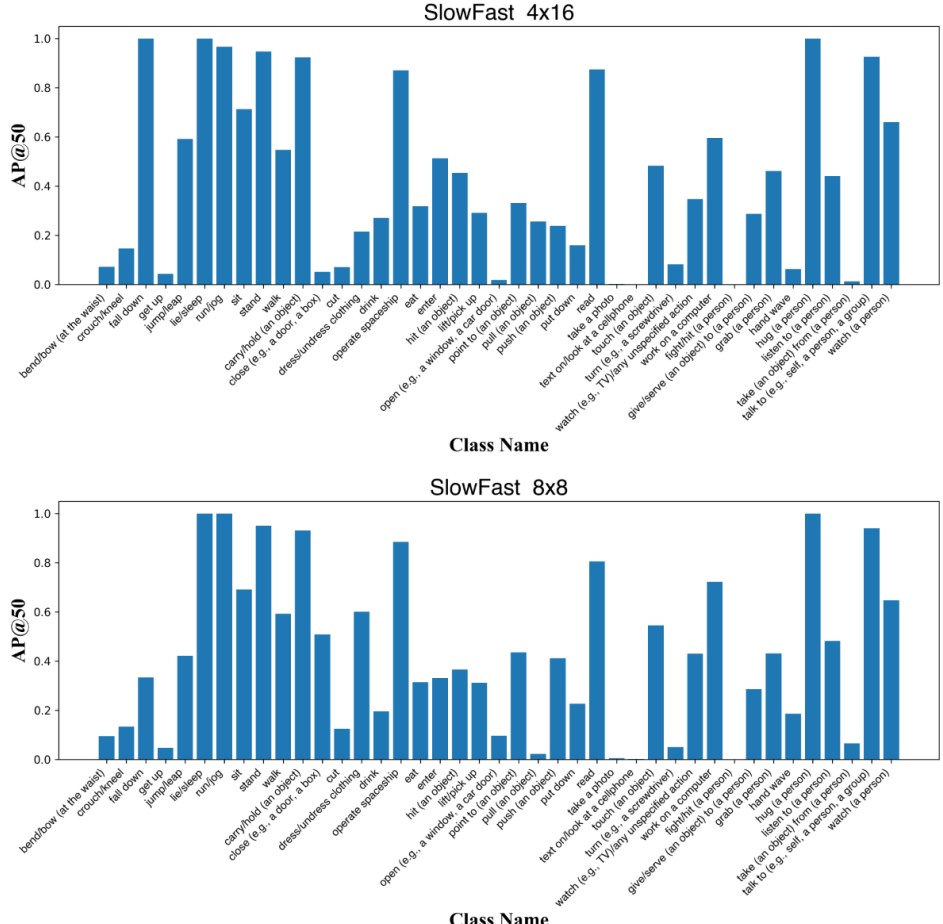

Figure 10: Per-class AP (%) on the MicroG-4M test set for **SlowFast** Feichtenhofer et al. (2019). Training regime: MicroG-4M fine-tuning. AP@0.5 over the 50 MicroG-4M action classes; class order matches Table 9. Legend uses $L \times S$ to denote clip length $\times$ sampling stride; settings match Table 1.

Table 11: HAR performance of Gemini 2.5 Pro on the MicroG-4M test set, evaluated on the 50-way multi-label action space with macro-averaging over classes.

| Model | Test Result | | | |
|---|---|---|---|---|
| | mAP (%) | F1-score (%) | Recall (%) | AUROC (%) |
| Gemini 2.5 Pro Comanici et al. (2025) | 14.07 | 25.44 | 46.41 | 71.43 |

of action IDs that are clearly performed by the highlighted person. We encode both ground-truth annotations and Gemini predictions as 50-dimensional $\{0, 1\}$ indicator vectors over the action label space and compute macro-averaged metrics over classes. Since Gemini does not expose calibrated per-class probabilities in this setting, the binary indicators are used both as decisions (for F1-score and recall) and as surrogate scores (for mAP and AUROC), and the ranking metrics should therefore be interpreted as conservative lower bounds. The resulting global performance is summarized in Table 11.

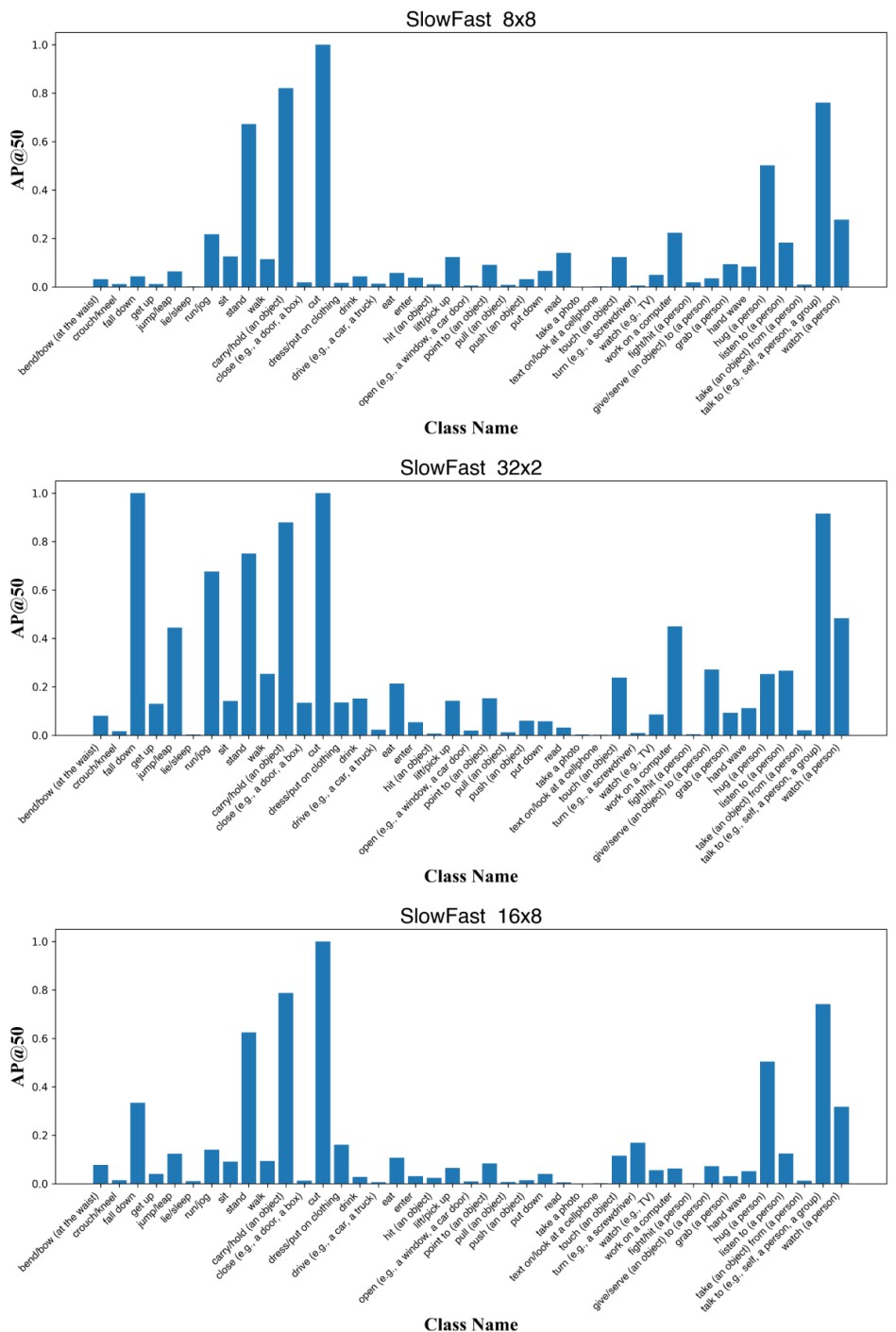

Figure 11: Per-class AP (%) on the MicroG-4M test set for **SlowFast** Feichtenhofer et al. (2019). Training regime: AVA fine-tuning with zero-shot evaluation on MicroG-4M. AP@0.5 over the 50 AVA-derived action classes. Legend uses $L \times S$ to denote clip length $\times$ sampling stride; settings match Table 1.

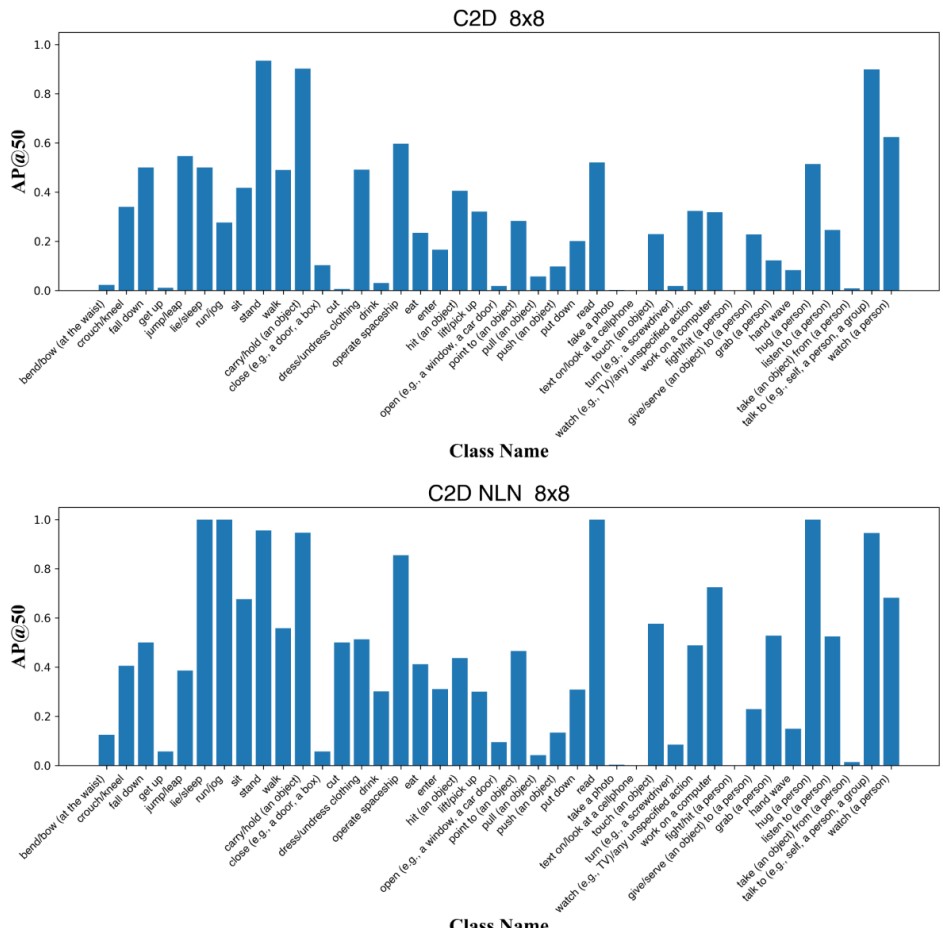

Figure 12: Per-class AP (%) on the MicroG-4M test set for **C2D** Feichtenhofer et al. (2019). NLN denotes the non-local augmentation of the backbone. Training regime: MicroG-4M fine-tuning. AP@0.5 over the 50 MicroG-4M action classes; class order matches Table 9. Legend uses $L \times S$ to denote clip length $\times$ sampling stride; settings match Table 1.

### G.5.1 Implementation Details

The evaluation procedure for the Human Action Recognition (HAR) task consists of three distinct stages, detailed below.

**Per-person Clip Generation.** The process begins by leveraging the provided spatio-temporal annotations, which specify the bounding box coordinates, frame index, and unique identifier for each person across the video dataset. For every unique video–person pair, a corresponding short video clip is constructed. Frames are decoded from the raw video files, and the target actor's bounding box in each frame is highlighted and annotated with the person identifier. These localized per-person clips are generated to ensure that the model consistently focuses its analysis on a single actor within the video.

**Gemini 2.5 Pro Inference.** The HAR label space encompasses 50 action classes. Each generated clip is uploaded to Gemini 2.5 Pro, and the model is prompted with a predefined text containing the list of all 50 action IDs and their descriptions. The instruction is to return a JSON list of the action IDs clearly performed by the highlighted subject. The model's textual output is subsequently parsed into a set of predicted action identifiers, with invalid or out-of-vocabulary labels being rigorously filtered.

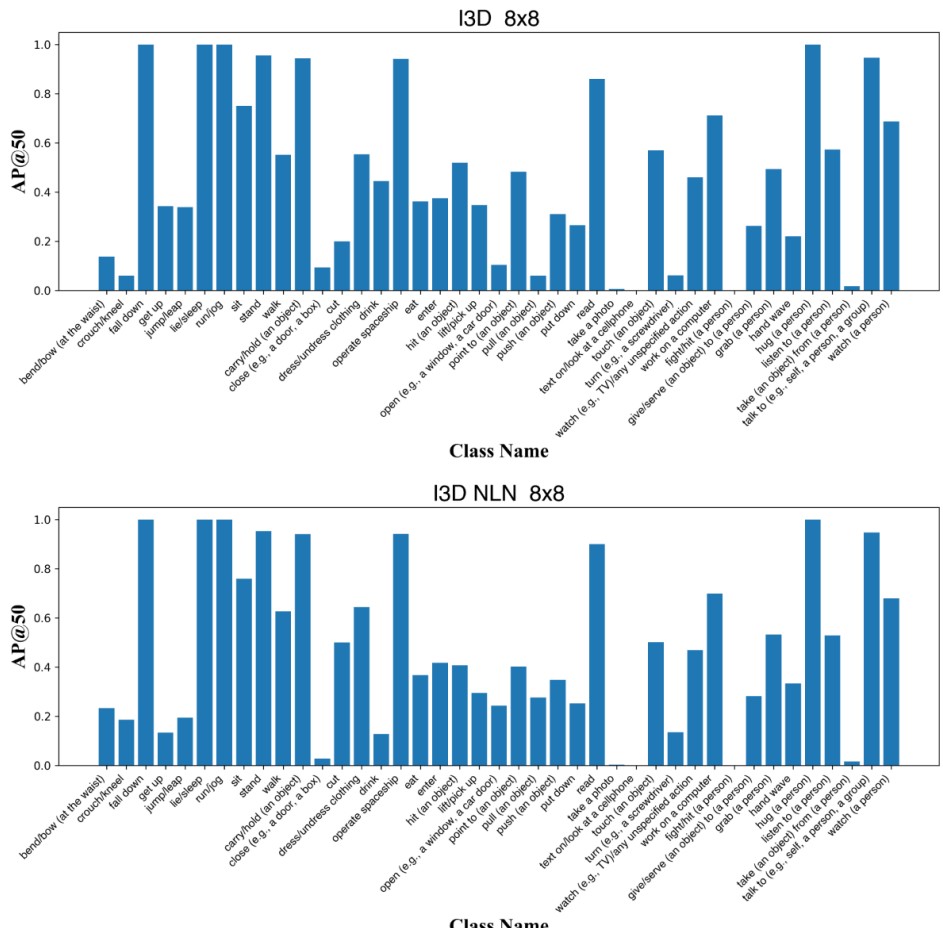

Figure 13: Per-class AP (%) on the MicroG-4M test set for **I3D** Carreira & Zisserman (2017). NLN denotes the non-local augmentation of the backbone. Training regime: MicroG-4M fine-tuning. AP@0.5 over the 50 MicroG-4M action classes; class order matches Table 9. Legend uses $L \times S$ to denote clip length $\times$ sampling stride; settings match Table 1.

**Metric Computation from Binary Indicators.** Ground-truth HAR annotations are aggregated from the action label file, establishing a set of true labels for each video–person instance. Both the ground-truth and the predicted label sets are then converted into 50-dimensional binary indicator vectors. Since the inference configuration was constrained to return only action IDs (a list output), the Gemini model does not provide calibrated per-class prediction probabilities. Consequently, these $\{0, 1\}$ indicators are directly employed as the decision variables for computing F1-score and Recalls, then as surrogate scores for calculating mAP and AUROC. Macro-averaged metrics are obtained by averaging the respective scores over the 50 action classes. The global metrics reported in Table 11 are aggregated over the entire MicroG-4M test set.

**Observations.** Compared with dedicated video architectures trained on MicroG-4M (Table 1), Gemini 2.5 Pro achieves noticeably lower mAP and AUROC, while exhibiting relatively high recall, which indicates a tendency to over-predict actions. Together with the cross-domain transfer results in Tables 2 and 3, this highlights a substantial distribution gap between terrestrial benchmarks and microgravity videos. Both conventional video encoders and a state-of-the-art multimodal language model trained predominantly on terrestrial data struggle to bridge this gap on the HAR task, suggesting that microgravity-specific dynamics and context cannot be handled by generic action priors or textual knowledge alone.

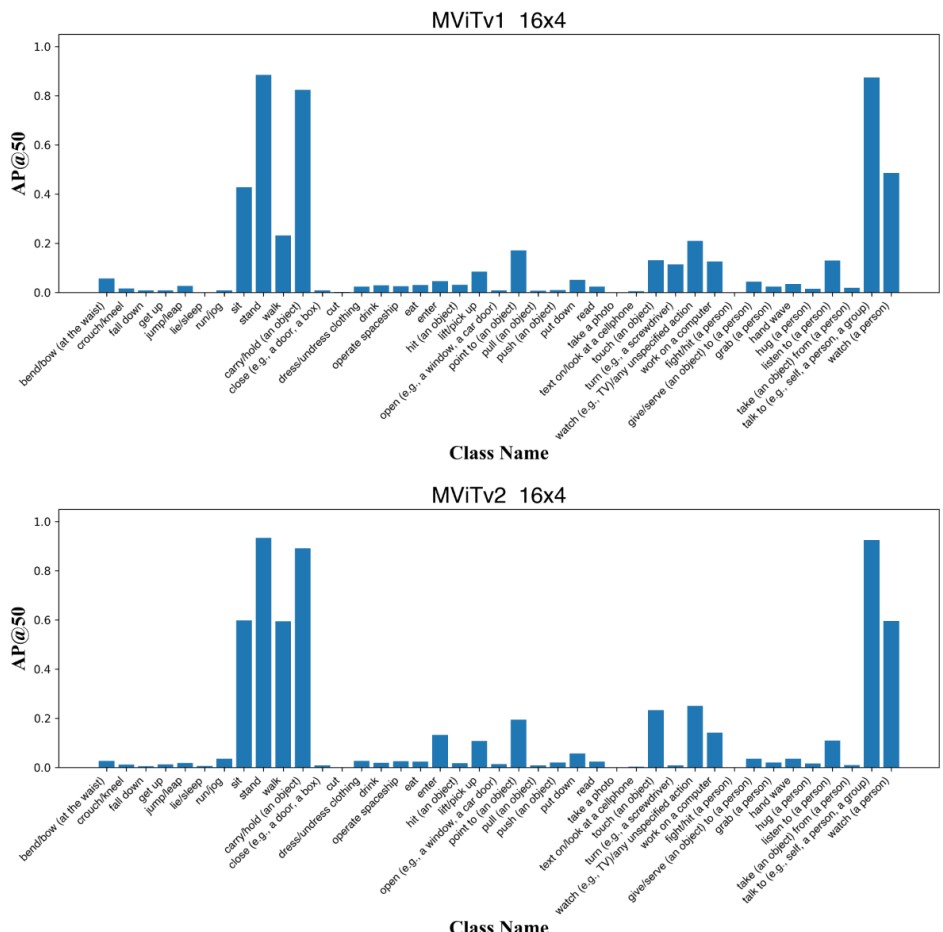

Figure 14: Per-class AP (%) on the MicroG-4M test set for **MViTv1** Fan et al. (2021) and **MViTv2** Li et al. (2022). Training regime: MicroG-4M fine-tuning. AP@0.5 over the 50 MicroG-4M action classes; class order matches Table 9. Legend uses $L \times S$ to denote clip length $\times$ sampling stride; settings match Table 1.

## H SUPPLEMENTARY FOR VIDEO CAPTIONING AND VISUAL QUESTION ANSWERING (VQA) TASK

### H.1 ADDITIONAL ANNOTATION EXAMPLES

Figure 16 illustrates two further representative examples of caption and VideoQA annotations in MicroG-4M. For each 3-second clip, annotators provide a dense caption that jointly encodes astronaut identity, cabin location, relevant equipment, and fine-grained hand–body–object interactions, followed by six visually grounded QA pairs that cover entity, attribute, action, spatial context, and temporal or causal reasoning, with an explicit *Not mentioned* option when appropriate.

### H.2 COMPUTATIONAL OVERHEAD

Table 12 reports warm-start latency and GPU memory usage across models. The observed efficiency differences are consistent with architectural factors such as video encoder capacity, tokenizer design, and the number of input frames, and can serve as a practical reference when choosing models under different computational budgets.

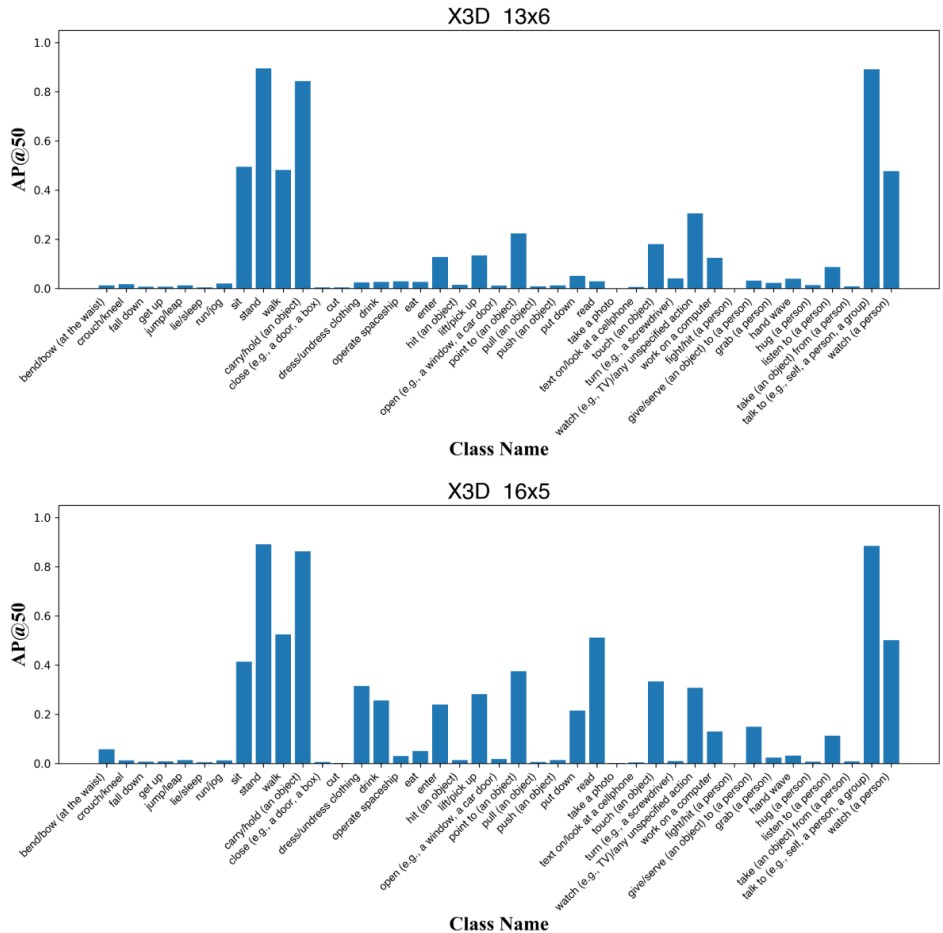

Figure 15: Per-class AP (%) on the MicroG-4M test set for **X3D** Feichtenhofer (2020). Training regime: MicroG-4M fine-tuning. AP@0.5 over the 50 MicroG-4M action classes; class order matches Table 9. Legend uses $L \times S$ to denote clip length $\times$ sampling stride; settings match Table 1.

Table 12: Computational overhead on MicroG-4M under warm-start conditions. We report median latency (p50) per clip and per sampled frame, together with peak GPU memory. #f denotes the number of input frames used during model inference.

| Model | Precision | #f | Caption | | VQA | | Peak GPU Mem (GiB) |
|---|---|---|---|---|---|---|---|
| | | | p50 (s/clip) | p50 (ms / sampled-frame) | p50 (s/clip) | p50 (ms / sampled-frame) | |
| Video-LLaVA Lin et al. (2024) | fp16 | 8 | 1.82 | 288.05 | 0.50 | 62.20 | 14.98 |
| Qwen2.5-VL Yang et al. (2024) | bf16 | 9 | 0.48 | 15.87 | 1.08 | 35.83 | 3.22 |
| Tarsier2-Recap-7B Yuan et al. (2025) | bf16 | 16 | 1.20 | 93.65 | 0.82 | 48.62 | 16.50 |

# I    PROMPT TEMPLATES FOR VIDEO CAPTION AND VQA ANNOTATION

This section provides the prompt templates used during caption refinement and VideoQA annotation. All prompts are applied after human annotators have written the initial captions and consulted the domain resources described in Section 3. In all cases, Multimodal Large Language Models (MLLMs) are instructed not to introduce new facts beyond what is visually or contextually supported.

## I.1    CAPTION REFINEMENT PROMPT

We use the following template to refine human-written captions, conditioning on four uniformly sampled keyframes while preserving all factual content.

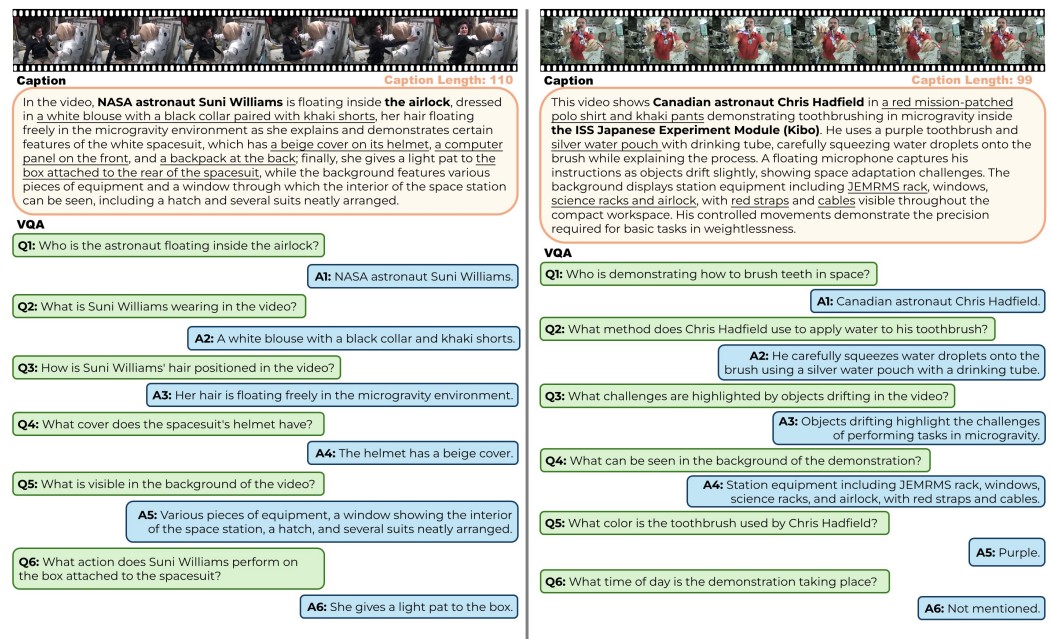

Figure 16: Examples of video caption and VQA annotations in MicroG-4M. Each panel shows a short clip, its corresponding caption, and six associated QA pairs, illustrating the level of detail and reasoning coverage used in our annotations.

```
[Instruction]
You are assisting with caption refinement for short astronaut videos.
You are given:
   (1) four keyframes sampled from a 3-second video clip, and
   (2) a human-written caption describing a 3-second video clip

Your task:
   - Use the visual evidence from the frames as the primary reference.
   - Improve grammar and fluency.
   - Increase clarity and readability.
   - Simplify overly complex sentences into a clear, concise style.
   - Optionally vary wording slightly to avoid repetition.
   - DO NOT add new facts or speculate beyond the caption and context.
   - DO NOT change any factual details.

Return a refined caption with similar length to the input, preserving all factual co

[Input frames]
Frame_1: {IMAGE_1}
Frame_2: {IMAGE_2}
Frame_3: {IMAGE_3}
Frame_4: {IMAGE_4}

[Input caption]
{CAPTION_TEXT}

[Context]
{CONTEXT_TEXT}

[Output]
Refined caption:
```

## I.2 VQA Candidate Generation Prompt

Given a refined caption and four keyframes from each 3-second clip, we generate diverse question–answer candidates that cover multiple reasoning dimensions. The following template is used:

```
[Instruction]
You are generating visual question-answer (VQA) pairs for a 3-second
astronaut video clip in microgravity.

You are given:
  (1) four keyframes representing the video, and
  (2) a refined caption that accurately summarizes the clip.

Treat the caption and keyframe images as the only reliable information
about the video.

Your tasks:
  1. Propose 8-10 diverse, natural, and meaningful Q&A pairs that are
     answerable from the visual content in the frames, supported by the
     caption.
     - Questions must be visually grounded in what can be seen.
  2. Cover a mix of:
     - Standard Wh-forms (who, what, where, how, when, why).
     - Foreground vs. background elements.
     - Coarse vs. fine-grained actions.
     - Identity, location, and equipment.
     - Temporal or causal relations.
     - At least one question that cannot be answered from the frames
       and caption; its answer must be exactly "Not mentioned".
  3. Use varied question types (Who / What / Where / When / Why / How)
     and avoid repeating the same content.
  4. DO NOT rely on sound, speech content, or subtitles.
  5. DO NOT rely on internal intentions or emotions that are not visible.
  6. DO NOT rely on external world knowledge beyond what is evident in
     the frames and caption.

[Input frames]
Frame_1: {IMAGE_1}
Frame_2: {IMAGE_2}
Frame_3: {IMAGE_3}
Frame_4: {IMAGE_4}

[Refined caption]
{CAPTION_TEXT}

[Output]
Return ONLY the selected Q&A pairs in the following format, with
no extra commentary:

Q: [question]?
A: [answer].
```

## I.3 VQA Filtering and Top-6 Selection Prompt

Candidate QA pairs are then filtered and ranked, and six diverse, visually grounded questions are selected.

```
[Instruction]
```

```
You are given a refined caption and a list of candidate question-answer
pairs for a 3-second astronaut video clip.

Your tasks:

  1. FILTER OUT any QA pairs that:
     - Rely on sound, speech content, or subtitles.
     - Require private intentions or emotions not visible in the clip.
     - Depend on external world knowledge not suggested by the caption.
     - Are duplicate or near-duplicate questions.

  2. RANK the remaining QA pairs by:
     - Logical consistency between question and answer.
     - Linguistic fluency and clarity.
     - Semantic relevance to the caption.
     - Informational value (how much useful visual information is tested).

  3. SELECT  6 final QA pairs that:
     - Cover diverse aspects (identity, action, body motion,
       spatial context, background, temporal/causal).
     - Are visually grounded in the clip.
     - Optionally include ONE unanswerable question, whose answer
       must be exactly "Not mentioned".

[Input]
Refined caption: {CAPTION_TEXT}
Candidate QA list: {JSON_LIST_OF_QA}

[Output]
Return a JSON list of the selected QA pairs in their final form,
each with:
Q: [question]?
A: [answer].
```

## J  ADDITIONAL DATASET VISUALIZATIONS

Figure 17 presents a collage of sampled frames to provide a quick visual impression of MicroG-4M, illustrating the dataset's core characteristics. The montage spans different crews, stations, and activities.

MicroG-4M's coverage is designed to approximate the breadth of everyday microgravity operations by including footage from both major contemporary orbital platforms and relevant training/simulation environments. Specifically, the dataset incorporates clips from orbital platforms, including various distinctive pressurized modules of the *International Space Station (ISS)* (e.g., Node modules, Cupola, Kibo/JEM, Columbus, Destiny/US Lab, BEAM) and the *Chinese Space Station (Tiangong)*. Beyond in-orbit footage, we include frames from simulated and conceptual microgravity contexts, such as parabolic flights and specialized film sequences, which broaden the scope of training and conceptual scenarios, particularly those used for extra-vehicular activity (EVA) preparation. This collection, sampled across different expedition periods and mission timelines, captures significant temporal and operational spread. Furthermore, the gallery highlights crew demographics and apparel consistent with documented daily life and work. Finally, the frames exhibit considerable variability, encompassing diverse illumination, camera viewpoints, aspect ratios, resolutions, group sizes, and motion types, covering both *intra-vehicular* tasks and *EVA-related* contexts.

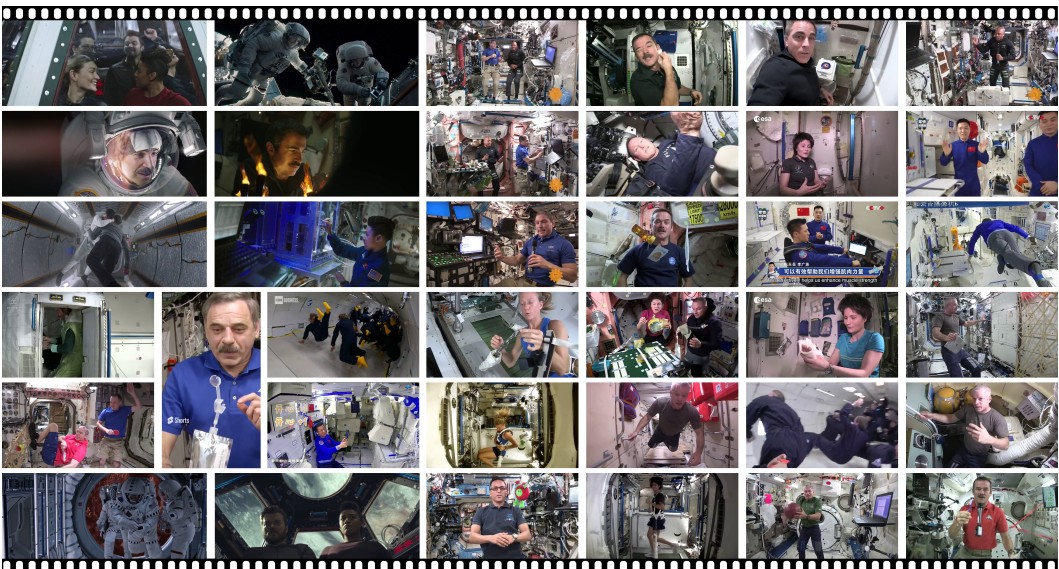

Figure 17: Representative frames from MicroG-4M.

