# OpenReview forum: "Go Beyond Earth: Understanding Human Actions and Scenes in Microgravity Environments"
_ICLR.cc/2026/Conference — ICLR 2026 Poster_

### Official Review · Reviewer_Dmaz · 2025-10-29

**Soundness:** 3
**Presentation:** 3
**Contribution:** 2
**Rating:** 6
**Confidence:** 3

**Summary:**

This paper proposes a new benchmark on video understanding.
The benchmark address the novel scenario of astronautic environment in space.
The benchmark contain 5k clips of short videos (3 seconds) with rich annotations.
This paper then benchmark state-of-the-art models using their dataset, results suggest the evaluated models do not work as good as they are in normal earth-based videos.

**Strengths:**

This paper addresses a novel scenario for video understanding.

The paper is well motivated.

The paper clearly documented the data collection pipeline.

The evaluation setting of baseline methods are also clearly written.

**Weaknesses:**

The main issue with this paper is that its distinction from the Earth-Video benchmark is not clearly presented.
The reported performance indicates that the benchmark is difficult, but it is unclear whether any unique aspects of the astronautic videos contribute to this difficulty.
Currently, the presentation of experiment section 5 feels like “just another video benchmark”, as it does not provide much surprising findings.


Regarding the baselines of human action recognition task, I’m curious whether the authors have tried VLMs like Gemini 2.5 Pro? I feel advanced VLMs like Gemini 2.5 Pro may be able to answer this. It would be nice to have them.

**Questions:**

This paper is clearly written, thus I don’t have questions regarding clarity.

I expect more qualitative results like Figure 1, 3 and 4.

The videos.zip in the supplementary does not work for me.

---

> ### Author Response · Authors · 2025-11-25
> **Author Response to Reviewer Dmaz - Part I**
>
> W1. Clarification on the gap between Earth and space video performance
>
> We thank the reviewer for highlighting the need to make the Earth–space distinction explicit. **Appendix E.1 now includes a more detailed per-class AP analysis under both training regimes (MicroG→MicroG and AVA→MicroG) across all backbones (C2D, I3D, Slow, SlowFast, X3D, MViTv1/v2, with and without non-local), with results summarized in Figs. 8 and 9.** Two *consistent signatures* show that the difficulty is specific to microgravity rather than just another hard benchmark.
>
> - Gravity-dependent Person Movement is fragile.
> Classes that presuppose an upright axis or rapid postural change remain near zero across models, including *bend/bow (at the waist)*, *crouch/kneel*, *get up*, *jump/leap*, lie/sleep, and run/jog. Typical confusions match this factor, for example sit vs bend/bow and stand or sit vs lie/sleep.
> - Transient force direction and short contact in Object Manipulation are hard, while device-anchored operations are stable. AP is low for *push (an object)*, *pull (an object)*, *put down*, *touch (an object)*, and *point to (an object)*. In contrast, device-centric actions with strong visual anchors are comparatively stable, for example read and work on a computer. carry/hold (an object) and lift/pick up are stronger than other manipulation classes, consistent with sustained hand–object contact cues.
>
> These class-level patterns, replicated across architectures in Figs. 8 and 9, indicate that the observed gap is driven by the loss of gravity-aligned priors and by force-direction or transient-contact ambiguities intrinsic to space footage, not by training recipes. We reference these findings in the main text and the detailed plots are provided in the appendix G for transparency.
>
> W2. On using VLMs like Gemini 2.5 Pro as HAR baselines.
>
> Thank you for the suggestion. Our benchmark treats HAR as supervised multi-label recognition with calibrated per-class scores and spatio-temporal localization. Closed-source VLMs such as Gemini 2.5 Pro do not output calibrated logits for the 50 actions or detection boxes, and their undisclosed pretraining data may include space footage. Including Gemini 2.5 Pro in the main HAR tables would therefore be unfair and not directly comparable.
>
> **We nevertheless provide a diagnostic HAR evaluation in the appendix G.5.** Using person-centric clips cropped from our boxes, we present a forced-choice list of the 50 actions and request a JSON list of predicted action IDs for the highlighted person. We compute macro mAP, AUROC, F1, and Recall by treating the returned list as binary scores per class. Because this protocol lacks calibrated confidences and boxes, we keep Gemini 2.5 Pro out of the main HAR leaderboards.
>
> **Importantly, we have added Gemini 2.5 Pro to the captioning and VQA results in the revised paper (see Table 4 and Table 5)**, and we report CIDEr, BLEU, ROUGE, METEOR, BERTScore, and the answer-semantic score. Please see the revised captioning and VQA tables for these numbers.
>
> Q1. Response on qualitative results.
>
> Thank you for the suggestion. In the revised paper, we have significantly expanded the qualitative results for better interpretation. Specifically, **Appendix E.1 now provides an enlarged set of success and failure cases for the HAR task**, extending the examples previously shown in Figures 3 and 4 (Figure 4 and 5 in revised PDF). Furthermore, **Appendix H offers expanded ground truth examples** for VQA and Video Captioning, aligning with the pattern analysis presented in Figure 1.

---

> ### Author Response · Authors · 2025-11-25
> **Author Response to Reviewer Dmaz - Part II**
>
> Q2. On the “videos.zip”.
>
> We appreciate the reviewer flagging this. The non-functional videos.zip placeholder was inadvertently left in the anonymized repository during review, and we cannot modify the anonymous package at this stage. In the final public repository, we will remove this placeholder and direct users to the annotation index with clear retrieval instructions.
>
> For copyright and platform-policy reasons, we do not redistribute third-party video files. This follows standard practice in action-recognition benchmarks, where datasets release annotations and source identifiers rather than re-hosting media. For example, AVA is distributed via annotations and source URLs, and Kinetics provides lists of source links and time spans instead of mirrored videos.
>
> In our release, files under *annotation_files* include a *video_id* that corresponds to the source platform identifier or the source film title, enabling readers to inspect content on original platforms or obtain access through legitimate channels. We do not endorse or facilitate unauthorized downloading, consistent with the terms of service of major streaming platforms, which generally restrict downloading or re-hosting content without an official download option.
>
> **To still give reviewers a concrete sense of the data, Appendix J now includes representative frame montages/screenshots from a subset of clips, illustrating typical scenes.** These examples are intended to provide visual intuition about the dataset while respecting copyright and platform policies.

---

> ### Comment · Reviewer_Dmaz · 2025-11-26
>
> I thank the authors for their response.
>
>
> ### W.1 Clarification on the gap between Earth and space video performance
>
> The authors included two new paragraphs in Line 955-965, unfortunately it is difficulty to understand which specific figure/examples are discussed by the first paragraph. This paragraph uses descriptive sentence, .e.g. "lift/pick up, put down, and hit (an object) are often confused with each other or with background motion". It is critical to point to corresponding visual examples.
>
> Similarly, the second paragraph Line 958-965 seems trying to provide quantitative analysis, but no tables is referred. It also introduces confusing terms, for example, "contextually well-anchored actions" is not explained.
>
> > Appendix E.1 now includes a more detailed per-class AP analysis under both training regimes (MicroG→MicroG and AVA→MicroG)
>
> Figure-8 presents 12 models and they are finetuned on MicroG and test on MicroG; Figure-9 presents only 2 models finetuned on AVA an test on MicroG. The number of models don't match (12 v.s. 2), and it is also very difficulty of compare across two figures — reader needs to scroll up and down.

---

> > ### Author Response · Authors · 2025-11-27
> > **Response to “Official Comment by Reviewer Dmaz”**
> >
> > We are grateful for the reviewer's engagement with the revised manuscript and appreciate the constructive focus on enhancing the presentation and clarity of our analysis. The comments on were particularly helpful.
> >
> > We have addressed these points comprehensively:
> >
> > W.1 Clarification on the gap between Earth and space video performance
> >
> > **Regarding difficulty in understanding which specific figure/examples are discussed (Lines 955-958) and the need for visual links:**
> >
> > We agree that explicit visual references are critical for the qualitative analysis.
> >
> > - **Improved Traceability (Qualitative):** We have **revised Appendix E.1 (formerly Lines 955-958)** to directly reference visual examples. The qualitative analysis is now explicitly tied to specific visual instances by including **clear panel and column references to Figure 5.** For example, the confusion between "*lift/pick up*, *put down*, and *hit (an object)*" is now linked to specific visual evidence, ensuring the descriptive sentences are empirically supported.
> >
> > **Regarding the second paragraph (Lines 958-965) lacking table references and introducing confusing terms:**
> >
> > We have restructured and clarified the **quantitative analysis** and refined the terminology.
> >
> > - **Improved Traceability (Quantitative):** The quantitative analysis, which was previously in Lines 958-965, has been **entirely relocated and integrated into Appendix G.3 (Per-Class AP Results)**. This analysis now explicitly references the corresponding per-class Average Precision (AP) figures.
> > - **Updated Terminology and Clarity**: We have refined the manuscript by **replacing the vague phrasing "contextually well-anchored actions."** This term was originally intended to categorize actions that leverage new, non-gravity-aligned strong visual anchors specific to the space environment after fine-tuning. These actions are primarily **goal-oriented human-object interactions** (e.g., *cut*, *eat*, *read*, *work on a computer*) which provide stable visual cues. Recognizing that this terminology was insufficiently precise for a technical report, we have replaced this phrasing. We have now **entirely rewritten Appendix G.3** to align with the new figures and provide a much clearer and more precise per-class contrast, focusing directly on the **two primary categories of challenging actions: orientation-invariant postures and transient-contact manipulations across the training regimes.**
> >
> > **Regarding the imbalance of models presented in Figure 8 (12 models) versus Figure 9 (2 models) and the difficulty in comparison:**
> >
> > We have addressed both the imbalance in the regimes analyzed and the difficulty in visual comparison.
> >
> > - **Addressing Regime Imbalance:** To mitigate the perceived imbalance in our analysis of training regimes, we have **incorporated new results** for the AVA $\rightarrow$ MicroG-4M transfer and expanded the presentation of these findings in **Table 2** within the main text. This better reflects the performance under different transfer scenarios.
> > - **Enhanced Per-Class Analysis and Comparison:** The per-class AP plots in **Appendix G.3** have been substantially **reorganized into Figures 8-15** for better comparison:
> >
> >     - They now utilize a **single-column layout**, arranging the two training regimes for each backbone **adjacently**. This positioning enables direct comparison with minimal scrolling, substantially improving interpretability.
> >
> >     - Specifically, the original Figure 8 (MicroG-4M → MicroG-4M) has been **expanded and corresponds to the current Figures 8, 10, and 12-15**, while the newly added transfer results, Figure 9 (AVA → MicroG-4M), **correspond to the current Figures 9 and 11**.
> >
> > These edits collectively improve the traceability from text to figures and tables and substantially enhance the readability and interpretability of the observed Earth-to-space performance gap.

---

### Official Review · Reviewer_fpxM · 2025-10-30

**Soundness:** 3
**Presentation:** 2
**Contribution:** 3
**Rating:** 6
**Confidence:** 4

**Summary:**

The paper introduces MicroG-4M, the first benchmark dataset designed for spatio-temporal and semantic understanding of human activities in microgravity environments. It addresses a critical gap in current vision research, as most existing datasets for video captioning and action recognition are recorded on Earth under normal gravity, whereas microgravity significantly alters human motion, interactions, and visual semantics. Constructed from real-world space mission footage and cinematic simulations, MicroG-4M contains 4,759 3 second video clips at 30fps covering 50 actions, 1,238 captions, and over 7,000 question–answer pairs centered on astronaut activities and scene understanding. The dataset supports three core tasks: fine-grained multi-label action recognition, temporal video captioning, and visual question answering.

**Strengths:**

1.  The contribution of MicroG-4M is highly interesting and original, as it introduces the first large-scale benchmark for understanding human actions, captions, and question answering in microgravity environments. The dataset offers significant potential for advancing research in vision-language modeling, domain adaptation, and embodied AI under extreme physical conditions.

2. The strength of this paper lies in the detailed presentation of dataset statistics and discussion with limitation. The authors provide a clear breakdown of the distribution across broad action types, the number of persons per clip, and fine-grained action frequencies, highlighting important patterns such as the dominance of single-person clips and the long-tail distribution of actions. Additionally, the per-class AP results convincingly demonstrate the value of MicroG-4M for microgravity-specific action recognition, while the high-density, VQA annotations further support rich semantic understanding.

**Weaknesses:**

1. In the collection methodology section, the author’s writing style is precise, formal, and research-oriented, making it suited for submission in conference. However, it leans toward being dense and information-heavy, which could benefit from slight simplification or the inclusion of visual aids (such as tables or flow diagrams) to enhance clarity and readability.

2. The author does not explicitly define the categories “Object Manipulation,” “Person Interaction,” and “Person Movement.” Instead, it appears that the reader is expected to infer their meaning from common sense or from the constituent fine-grained actions. It would improve clarity if the author briefly described each category with examples. For instance, "Object Manipulation could be defined as actions where a person interacts with objects in the environment, including picking up, carrying, holding, pushing, pulling, operating equipment, or using tools. Examples include carry/hold object, push object, operate spaceship, or using a computer. In microgravity, these actions are particularly important because the dynamics of motion and object handling differ from those on Earth."

3. While the authors state that all captions and QA annotations were created manually by annotators with “domain guidance,” they do not specify what type of guidance was provided or give concrete examples. It would be helpful to clarify whether this guidance included official space agency documents, astronaut manuals, mission reports, or expert review, and to briefly describe what information from these sources was used (e.g., spacecraft layout, standard operating procedures, or typical astronaut activities). Providing such details would improve transparency and help readers better assess the quality and reliability of the annotations.

**Questions:**

1. For video captioning and VQA, the authors mention using large language models (LLMs) but do not describe the prompts or prompting strategy employed. Providing information about the prompts, including their format, instructions, or examples, would improve reproducibility and allow readers to better understand how LLMs contributed to annotation quality.

2. It is unclear how video input is processed to generate captions. Do the authors first extract keyframes, or do they read video frames sequentially? If keyframes are used, the authors should specify the algorithm or criteria for keyframe selection. Additionally, the caption generation process using Visual-Language Models (VLMs) or Multimodal Large Language Model (MLLM) is not described in detail, clarifying whether captions are generated per frame, per keyframe, or for the entire clip would improve reproducibility.

3. Employing VLMs may introduce computational overhead, especially for high-resolution frames or sequential multi-frame processing. For practical deployment, it would be helpful if the authors reported the hardware configuration (GPU type, CPU, RAM), memory usage, and average execution time per frame or video clip. This information would provide readers with a clearer understanding of the method’s efficiency and scalability.

4. The authors state that each three-second clip contains six QA pairs; however, it is unclear how these questions were generated or selected. It would be helpful to clarify whether the questions were derived directly from the video content or generated based on the captions. The paper should also describe the question selection process, including examples of question types and how balance was maintained across reasoning categories such as temporal reasoning. The author could explain such as Six QA pairs covering different aspects: (1) Identity (“Who is the astronaut?”), (2) Action detail (“What does she do with her hands?”), (3) Body motion (“How do her head and shoulders move?”), (4) Spatial context (“What items can be seen behind?”), (5) Static background recognition (“What remains stationary?”), (6) Unanswerable / implicit reasoning. However, the authors do not indicate whether such systematic consideration guided their question design. A clearer explanation of the QA generation and selection methodology would greatly enhance reproducibility and transparency.

---

> ### Author Response · Authors · 2025-11-25
> **Author Response to Reviewer fpxM - Part I**
>
> Thank you for the constructive comments. We address each point and indicate the exact clarifications we added in the revision.
>
> W1. Presentation and visual clarification.
>
> We thank the reviewer for the suggestion. To improve clarity without reducing detail, we have **added a visual diagram summarizing the full data collection and annotation pipeline in Figure 2**. This supplements Section 3 by providing a structured overview of all stages including clip selection, human filtering, caption annotation, VQA generation, and refinement. We believe this improves readability while preserving the technical completeness of the methodology.
>
> W2. Category definitions: OM, PI, PM
>
> We thank the reviewer for noting this. In the revised paper, we have **added concise definitions of Object Manipulation (OM), Person Interaction (PI), and Person Movement (PM) in Section 4**. These three macro-categories are **directly derived** from AVA’s atomic actions. OM refers to object- or equipment-centered actions (e.g., push object, work on computer), PI to person-to-person interaction (e.g., talk to person, hug), and PM to posture or motion of the individual (e.g., walk, bend, sit). Same macro-categorizations are *widely adopted* in related fine-grained action recognition works based on AVA, such as [1], to facilitate clearer performance analysis. This improves *interpretability* of the per-category statistics while maintaining full label *compatibility* with AVA.
>
> [1] Peng, Kunyu, et al. "Referring atomic video action recognition." European Conference on Computer Vision. Cham: Springer Nature Switzerland, 2024.

---

> ### Author Response · Authors · 2025-11-25
> **Author Response to Reviewer fpxM - Part II**
>
> W3/Q1/Q4. Clarification on domain guidance and annotation reliability
>
> We appreciate the reviewer’s request for clarification on the types of domain guidance used during annotation. Below we provide concrete details on both the sources referenced and the roles they played in ensuring the accuracy and reliability of our captions and VQA.
>
> **Domain resources used.**
> During annotation, all annotators had access to a curated repository of publicly available, authoritative resources, including:
>
> * Official crew rosters and astronaut biography databases (e.g., NASA, ESA, JAXA)
> * Historical spacecraft module layout diagrams (e.g., Destiny Lab, Kibo, Columbus)
> * Equipment specification documents (e.g., SAFER, airlock controls)
> * Mission operation reports and task logs (e.g., EVA sequences, docking events)
> * Onboard video transcripts and official archival footage
>
> These references were used systematically to resolve ambiguities in astronaut identity, cabin context, equipment usage, and to ensure correct naming conventions across modules and objects.
>
> **Caption and VQA annotation**
> Thank you for this suggestion. In the revised submission, we have **added a new appendix section I** that provides the exact prompt templates used during annotation. These include both (i) the LLM-based refinement stage for caption writing, which focuses on **improving grammar, clarity, lexical diversity, and simplifying sentence structure**, and (ii) the **multi-stage generation and filtering process** for VQA question–answer pairs. We also added references to this appendix in Section 3 to ensure that the human–LLM annotation workflow is fully transparent and reproducible.
>
> The captions were written by human annotators based on frame-by-frame visual inspection. They include not only the core action and object-level description, but also detailed references to:
> - Astronaut identity
> - Cabin location and background spatial layout
> - Subtle hand-body-object interactions
> - Equipment appearance and specific usage
> - Astronaut posture, orientation, and visual traits (e.g., helmet on/off, suit worn)
>
> This information was *cross-checked* against the above *domain resources* to ensure semantic accuracy. LLMs were **only** used after human writing to improve grammatical fluency and lexical richness, with all final captions validated by annotators.
>
>
> VQA generation follows a controlled **two-stage process**. First, diverse questions are generated based on each refined caption, using prompts designed to elicit questions that vary across:
>
> we clarify that each 3-second clip is annotated with six QA pairs, generated through a structured caption-grounded process. First, candidate question–answer pairs are generated from each refined caption using prompt templates designed to elicit variation across multiple reasoning dimensions:
>
> - Wh-forms (Who / What / Where / When / Why / How)
> - Foreground vs. background elements
> - Coarse vs. fine-grained actions
> - Identity, location, and equipment
> - Temporal and causal relations
> - One deliberately unanswerable question per set
>
> A refinement pipeline then *filters out* candidates requiring sound, intent, or non-visual cues. Remaining questions are ranked by the LLM based on logical consistency, fluency, semantic relevance, and information value. Annotators review the top-ranked QA pairs, eliminate hallucinations, revise prompts if needed, and verify that all retained pairs are visually grounded or explicitly marked as “Not mentioned” when unanswerable.
>
> This pipeline ensures that each final set of six QA pairs reflects semantic diversity, visual fidelity, and controlled reasoning scope. **We have added an appendix section I detailing the prompt templates and representative question types.**
>
> Q2/Q3. Computational overhead of VLMs
>
> Captions in our dataset are manually written by annotators after watching the full 3-second video clip at 30 fps. No keyframe extraction is used for generation. In a separate refinement step, we apply a VLM-based editing pipeline that takes the initial human-written caption and four uniformly sampled keyframes as input. This step improves fluency, factual correctness, expression diversity, and simplifies complex sentences into standard captioning style. Visual-language models (VLMs) and MLLMs are otherwise used only for evaluation, not for data generation. During benchmarking, each model receives video inputs according to its own default temporal sampling (see Tables 4 and 5). Any inference overhead reflects the model’s design and is not a constraint of the benchmark. **For reproducibility, we have added hardware specifications in Section 5. To further clarify the practical cost of using VLMs, Appendix H.2 now reports warm-start computational overhead on MicroG-4M, including median latency per clip and per sampled frame, as well as peak GPU memory usage for representive open-sourced evaluated models.**

---

### Official Review · Reviewer_baMz · 2025-11-01

**Soundness:** 3
**Presentation:** 3
**Contribution:** 2
**Rating:** 4
**Confidence:** 4

**Summary:**

This work introduces MicroG-4M, the first large-scale dataset specifically curated for human action recognition and vision-language understanding in microgravity environments. MicroG-4M includes 4,759 clips covering 50 actions, 1,238 context-rich captions, and over 7,000 question–answer pairs on astronaut activities and scene understanding. MicroG-4M aims to support three core tasks: fine-grained multi-label action recognition, temporal video captioning, and visual question answering, thereby enabling a comprehensive evaluation of both spatial localization and semantic reasoning in microgravity contexts.

**Strengths:**

1.The paper is well-written and easy to understand.

2.This work introduces MicroG-4M dataset, which is a valuable contribution to fill the gap of video understanding benchmarks under microgravity scenarios.

**Weaknesses:**

1.One concern for this work is the technical contribution. From my point of view, the major contribution of this work comes from its data collection and organization, while the methodological contributions are missing, i.e., there is no specifically designed baselines for video understanding under microgravity scenarios nor novel insights from this work. This decreases the overall contributions of this work.

**Questions:**

Please refer to the weaknesses.

**Details Of Ethics Concerns:**

N/A.

---

> ### Author Response · Authors · 2025-11-25
> **Author Response to Reviewer baMz**
>
> Thank you for the constructive comments that prompted a clearer articulation of the technical contribution, baselines, and insights.
>
> R1. “Technical contribution is missing; no specifically designed baselines or novel insights.”
>
> This paper is scoped as a **benchmark** for microgravity video understanding. It defines standardized tasks and metrics, specifies a fixed evaluation protocol, reports broad baselines across model families, and surfaces reproducible insights.
>
> **(i) Fixed cross-domain protocol that isolates gravity.**
> Section 5 uses a matched transfer setup. Models are initialized on Kinetics, fine tuned on AVA with aligned clip length, fps, frame sampling, optimizer, and schedules, then evaluated zero shot on MicroG-4M. In parallel, AVA to JHMDB serves as a terrestrial contrast. With recipes held constant, the consistent gap on MicroG-4M indicates a physics-driven shift caused by the absence of a stable gravity prior rather than implementation variance. Table 3 quantifies this, for example *SlowFast 32×2 reaches 47.50 mAP on JHMDB and 23.81 on MicroG-4M*. The protocol is specified in the benchmark and implemented in the anonymized repository for review.
>
> **(ii) Broad, neutral baselines across families and tasks.**
> For human action recognition (HAR), we evaluate representative CNN plus non-local and Transformer lines, including C2D, I3D, Slow, SlowFast, MViT, and X3D, under a unified setting. For captioning and VQA, we report strong open-source systems such as VideoChatGPT, mPLUG-Owl-3, LLaVA-Next, Video-LLaVA, Qwen2.5-VL, and InternVideo, together with closed-source systems such as GPT-4o and Gemini 1.5 Pro, in a single table. This provides a transparent yardstick without introducing a new architecture.
>
> **(iii) Actionable, testable insights surfaced by the benchmark.**
>
> *Ranking and window length.* On MicroG-4M, CNNs with non-local modules outperform pure ViTs under matched budgets, and longer temporal windows help when gravity-aligned motion consistency is reduced. See Table 1 and Section 5.2.
>
> *Physics-driven gap.* The AVA to MicroG drop, compared with AVA to JHMDB under identical recipes, points to gravity absence as the primary factor rather than training tricks. See Table 3.
>
> *Characteristic failure modes.* Earth-trained models confuse floating or inversion with sit or bend, and passive drift with carry or hold. See Section 5 and Appendix E.
>
> *Language evaluation signals.* Space-station descriptions are domain heavy and paraphrase rich. We therefore report n-gram metrics such as CIDEr, BLEU, ROUGE, and METEOR, together with embedding-based metrics such as Sentence-BERT, BERTScore, and an answer semantic score for VQA. This avoids over-penalizing correct paraphrases while still detecting content errors and hallucinations. See Tables 4 and 5.
>
> **(iv) Transparent data and annotation pipeline with domain guidance and ethics.**
> Captions and QA are written by *human annotators* with documented domain guidance. We compile public space-agency sources such as mission objectives, crew rosters, module layouts, and equipment maps to name astronauts, compartments, and devices correctly. Multimodal Large language models are used to generate and rank candidate questions and to improve grammatical fluency, while final selections and all domain facts are *verified by annotators*. For ethics and licensing, we release annotations and metadata with retrieval pointers rather than raw videos, together with an anonymized resource for audit and reproduction.
>
> **We added in the Introduction section to state the benchmark scope, protocol, breadth of baselines, and the key findings above.**
>
> **Contribution in Brief**
> This submission defines a standardized evaluation regime for microgravity, provides strong and neutral baselines across model families, and yields reproducible insights into model ranking changes, the physics-driven gap, and characteristic error structures. **MicroG-4M is intended as a benchmark**, consisting of 4,759 3-second video clips at 30fps covering 50 actions, 1,238 captions, and over 7,000 QA pairs, foundation on top of which future work can build microgravity-aware architectures and unified frameworks that jointly model fine-grained HAR and VideoQA under microgravity. These points are now stated explicitly in the revised manuscript so that the benchmark contribution is unmistakable.

---

### Author Response · Authors · 2025-11-25
**Global Response**

Dear Reviewers,

We are writing to express our sincere gratitude for the thoughtful, comprehensive, and highly constructive feedback on our submission. We deeply appreciate the time and effort you dedicated to reviewing our submission.

We have carefully considered each point raised and have provided detailed responses to all comments in our individual response document. A revised version of the manuscript, reflecting all requested changes, has been uploaded. For your convenience, all modifications in the text have been clearly highlighted in a blue color.

Thank you once again for your invaluable input, which has significantly enhanced the clarity and quality of our work.

Best regards,

The Authors of submission #16618

---

### Author Response · Authors · 2025-12-01
**Summarization for AC**

Dear AC,

Thank you for reviewing our paper and rebuttal. We summarize the paper’s main contributions and the key revisions.

## Contribution Summary

Our paper introduces **MicroG-4M**, a benchmark for human action recognition (HAR) and vision–language understanding in **microgravity**. It comprises a new dataset, a standardized evaluation regime and broad, neutral baselines:

-	**Dataset and Tasks**. MicroG-4M contains 4,759 clips with **50 AVA-compatible actions** and person boxes, **1,238 human-written, domain-guided captions**, and over **7,000 QA pairs**. It supports three tightly coupled tasks under a single benchmark: fine-grained multi-label HAR with spatio-temporal localization, temporal video captioning, and visual question answering (VQA) focused on astronaut activities.

-	**Standardized Cross-Domain Protocol**. All video models are initialized on Kinetics. We evaluate **in-domain performance** (fine-tuned/tested on MicroG-4M) and **cross-domain transfer** (AVA→MicroG zero-shot, using AVA→JHMDB as a terrestrial control). This separates microgravity effects from implementation choices.

-	**Broad Baselines and Ranking Changes**. For HAR we systematically evaluate representative backbones (C2D, I3D, Slow, SlowFast, X3D, MViTv1/v2, $\pm$ Non-local) under a unified setup. For captioning/VQA, we include open-source VLM/MLLMs (VideoChatGPT, mPLUG-Owl-3, LLaVA-Next, Video-LLaVA, Qwen2.5-VL, InternVideo, Tarsier2) alongside GPT-4o and Gemini 1.5/2.5 Pro. We observe that **CNNs with non-local modules tend to outperform pure ViTs at similar compute**, and **longer temporal windows** provide greater microgravity gains than on Earth benchmarks.

-	**Microgravity-Specific Error Structure**. Across architectures and training regimes, **gravity-dependent Person Movement** classes remain near zero AP and are frequently confused. **Transient-contact Object Manipulation** classes are systematically difficult, while **device-anchored actions** with strong visual anchors are stable.

-	**Transparent Annotation Pipeline**. Captions and QA are **human-written** using curated domain resources for fact verification. MLLMs are used only for grammar/style refinement. Like AVA/Kinetics, we will release **annotations and source identifiers**.

## Summary of Rebuttal and Revisions

Revisions focused on clarifying the technical role, improving transparency, and restructuring the Earth–space analysis:

- **Benchmarking Scope and Technical Contribution**: We explicitly define the core technical contribution as: (i) defining a **fixed cross-domain evaluation regime** that isolates gravity, (ii) providing **broad, neutral baselines** (**7** video backbones, **9** VLMs/MLLMs), and (iii) extracting **reproducible microgravity-specific insights** on performance gaps and failure modes.
- **Data and Annotation Clarity**: We added **Figure 2** summarizing the  pipeline and update definitions of **Object Manipulation (OM), Person Interaction (PI), Person Movement (PM)**. We detailed the **domain guidance** used by annotators.
- **Reproducible Pipeline**: The **exact prompt templates** for caption refinement and VQA generation are now given in appendix. The VQA pipeline is now fully described, clarifying how we obtain “six QA per clip” with controlled diversity and purely visual grounding.
- **Restructured Analysis**: The Earth–space performance analysis is clearer. Qualitative failure modes (**Appendix E.1**) now explicitly reference **Figure 5** panels. Quantitative per-class analysis is in **Appendix G.3**. Per-class AP figures (**Figures 8–15**) were **reorganized into a single-column layout**, placing **MicroG→MicroG** and **AVA→MicroG** adjacently for easier comparison. AVA→MicroG transfer results are expanded in a main-text table.
- **VLM/MLLM Role and Overhead**: We emphasized MLLMs are not label generators but only for caption refinement and as baselines in VQA. **Appendix H.2** now reports median latency and peak GPU memory usage for representative models. We included a **diagnostic HAR experiment** and integrated **Gemini 2.5 Pro** results into the captioning/VQA tables.
- **Presentation and Accessibility**. We expanded qualitative HAR and VQA/captioning examples in the appendices to complement the aggregate metrics and patterns discussed in the main text. We clarified the final release plan: like AVA and Kinetics, we will provide **annotations plus video identifiers**, and additionally offer frame montages to give visual intuition.

## Overall
Overall, the revision makes the benchmark’s role as a **standardized evaluation regime for microgravity video understanding** explicit, strengthens the methodological and annotation descriptions, and substantially improves the readability of the Earth–space analysis. We hope this summary helps convey how MicroG-4M goes beyond a dataset release to provide a reproducible protocol and insight framework for studying human activities and vision–language models in **microgravity**.

---

### Meta-Review · Area_Chair_uCXv · 2026-01-05

**Summary:**

The paper proposes MicroG-4M that is a benchmark for human action recognition and vision–language understanding in microgravity. Two reviewers are positive (they scored 6) about this work while one (with score 4) raised concern about the lack of the methodological contributions. After reading the reviews and rebuttal, the AC believes that the paper looks interesting and important not because it targets space alone but because it exposes fundamental weaknesses in video understanding models when core physical assumptions are violated.

**Reviewer Concerns:**

It seems all concerns have been addressed by the rebuttal.

**Reviewer Scores:**

Probably all agree on score 6

---

### Decision · Program_Chairs · 2026-01-26

Accept (Poster)